# Improving Set Function Approximation with Quasi-Arithmetic Neural Networks

**Tomas Tokar**
University of Toronto
Wondeur AI
tomas@wondeur.ai

**Scott Sanner**
University of Toronto
Vector Institute
ssanner@mie.utoronto.ca

## Abstract

Sets represent a fundamental abstraction across many types of data. To handle the unordered nature of set-structured data, models such as DeepSets and Point-Net rely on fixed, non-learnable pooling operations (e.g., sum or max) – a design choice that can hinder the transferability of learned embeddings and limits model expressivity. More recently, learnable aggregation functions have been proposed as more expressive alternatives. In this work, we advance this line of research by introducing the Neuralized Kolmogorov Mean (NKM) – a novel, trainable framework for learning a generalized measure of central tendency through an invertible neural function. We further propose quasi-arithmetic neural networks (QUANNs), which incorporate the NKM as a learnable aggregation function. We provide a theoretical analysis showing that, QUANNs are universal approximators for a broad class of common set-function decompositions and, thanks to their invertible neural components, learn more structured latent representations. Empirically, QUANNs outperform state-of-the-art baselines across diverse benchmarks, while learning embeddings that transfer effectively even to tasks that do not involve sets.

## 1 Introduction

Sets represent a fundamental abstraction for various data types, examples of such include, set of objects placed in a scene (Eslami et al., 2016; Kosiorek et al., 2018), point clouds in a physical space (Chang et al., 2015; Wu et al., 2015), or a set of reinforcement learning agents (Sunehag et al., 2017). Developing neural methods that can effectively learn to approximate set functions is therefore crucial for advancing machine learning applications across diverse domains.

Sets impose no inherent ordering, making the design of neural architectures for processing them a unique challenge. To properly handle this property, neural architectures must exhibit permutation invariance, i.e., ensure that the output remains unchanged regardless of the order of the input elements. To guarantee this, most existing approaches such as DeepSets (Zaheer et al., 2017) and PointNet (Qi et al., 2017a) rely on pooling operations (e.g. sum, or max) that aggregate embeddings of the individual elements or their combinations into a single fixed-size representation, which is in turn processed to produce the final prediction (Murphy et al., 2019) (cf. Figure 1).

Because the pooling operation is fixed and non-trainable, the approximation burden is thus divided between the only two neural components: an *encoder* $\phi$, which maps each set element (or their combinations) to a latent representation, and an *estimator* $\rho$, which processes the pooled representation to produce the final prediction. Therefore, the encoder is forced to learn embeddings that are tailored not only to the downstream task but also to the *a priori* chosen pooling operation, restricting its ability to learn a more general or transferable latent representations (Soelch et al., 2019; Wagstaff et al., 2019; Bueno & Hylton, 2021; Wagstaff et al., 2022).

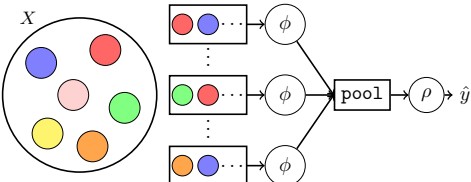

Figure 1: Generalized framework for set function learning, involving encoder $\phi$, estimator $\rho$ and pooling operation (e.g. sum, or max).

More recently, learnable aggregation functions have been proposed as more expressive alternatives (Locatello et al., 2020; Pellegrini et al., 2021; Kimura et al., 2024). To advance this line of research, we introduce a trainable modification of the *Kolmogorov mean* (a.k.a. quasi-arithmetic mean) (Bullen & Bullen, 2003), an important generalization of means that encompasses many commonly used measures of central tendency (e.g. arithmetic, geometric, harmonic mean, *etc.*). The Kolmogorov mean induces set aggregation via a selection of invertible *generating function*, which we implement using an invertible neural network, resulting in highly expressive learnable pooling operation. The use of trainable Kolmogorov mean enables more principled set-function approximation, leading to more structured latent spaces, improved encoder transferability, and consistently better downstream performance.

**Main contributions**   (i) We propose the first machine-learnable modification of the Kolmogorov mean. (ii) We propose quasi-arithmetic neural networks (QUANNs), which adopt the newly introduced neuralized Kolmogorov mean to serve as a learnable aggregation function. (iii) We support our proposal with theoretical and empirical analysis, demonstrating that QUANNs: (a) can approximate mean- (in the generalized sense) and, under mild assumptions, also max-decomposable set functions arbitrarily well; (b) QUANNs enable learning of more structured embeddings and more transferable encoders than existing models; and (c) overall, they outperform state-of-the-art (SOTA) set function models.

## 2   PRELIMINARIES

**Set-structured data**   Let $\mathcal{X}$ denote the input space, and let $\mathcal{D} = \{(\mathbf{X}_i, y_i)\}_{i=1}^{N}$ be a dataset where each $\mathbf{X}_i$ is a finite set of elements: $\mathbf{X}_i = \{\mathbf{x}_{i,1}, \ldots \mathbf{x}_{i,n_i}\}$, and $y_i$ is the corresponding label. For simplicity, we assume that each $\mathbf{X}_i$ is a homogeneous set, i.e., all elements of $\mathbf{X}_i$ come from the same domain $\mathcal{X}$: $\mathbf{x}_{ij} \in \mathcal{X}$. However, our framework can be naturally extended to heterogeneous sets. Additionally, we restrict our attention to the countable case, where all sets $\mathbf{X}_i$ are finite and of arbitrary cardinality $n_i$ that varies across the inputs.

**Set function modeling**   We seek to learn a set function $F : \mathcal{P}_f(\mathcal{X}) \to \mathcal{Y}$ where $\mathcal{P}_f(\mathcal{X})$ denotes the set of all finite subsets of $\mathcal{X}$, and the output space $\mathcal{Y}$, which is usually either $\mathbb{R}$ (scalar case, e.g. regression), or $\mathbb{R}^{d_{\text{out}}}$ for some fixed output dimensionality $d_{\text{out}} \in \mathcal{N}$ (in the vector-valued prediction case, e.g. classification). Our objective is to approximate the function $F$ using a neural network, trained on the dataset $\mathcal{D}$ by minimizing a loss function $\mathcal{L}$ obtained from the predicted outputs $\hat{F}(\mathbf{X}_i)$ and the labels $y_i \in \mathcal{Y}$.

**Permutation invariance**   A key requirement for set function modeling is *permutation invariance*, i.e., the output of the set function should remain unchanged under any reordering of the elements in the input set. Formally, we require the target function estimate $\hat{F}$ to satisfy:

$$\hat{F}(\mathbf{X}) = \hat{F}(\pi(\mathbf{X})) \quad \forall \mathbf{X} \subseteq \mathcal{X}, \ \forall \pi \in \mathcal{S}_{|\mathbf{X}|}, \tag{1}$$

where $\mathcal{S}_{|\mathbf{X}|}$ denotes the symmetric group of all permutations over the elements of the set $\mathbf{X}$, and $\pi(\mathbf{X})$ is the permutation of $\mathbf{X}$ under $\pi$.

## 3   RELATED WORK

### 3.1   JANOSSY POOLING

Most current methods for learning set functions can be viewed as specific instances of a broader generalization referred to as *Janossy pooling* (Murphy et al., 2019), which can be formalized as follows:

$$\hat{F}(\mathbf{X}) = \rho \left( \text{pool}_{\pi \in \mathcal{P}_k(\mathbf{X})} \phi \left( \pi(\mathbf{X}) \right) \right) \tag{2}$$

where: $\mathcal{P}_k(\mathbf{X})$ indicates all $k$-permutations of a set $\mathbf{X}$; *encoder* $\phi$ is a neural function that learns latent embedding of the input permutations; pool is a permutation-invariant *pooling operation* that aggregates the obtained embeddings, and *estimator* $\rho$ is another neural function that maps an aggregated embeddings to the final output.

Table 1: Unified comparison of the existing methods of Janossy pooling and QUANNs (ours); $\phi$, $\rho$, and $\psi$ are neural functions; $w$ indicates trainable parameter; $P_2(\mathbf{X})$ indicates all 2-permutations (pairs) of the input set and $n$ is set cardinality.

|  | MODEL | FUNCTIONAL FORM | REF. |
|---|---|---|---|
| **Unary** | DeepSets | $\rho\left(\sum_{i=1}^{n} \phi(\mathbf{x}_i)\right)$ | Zaheer et al. (2017) |
|  | PointNet | $\rho\left(\max_{i=1}^{n} \phi(\mathbf{x}_i)\right)$ | Qi et al. (2017a) |
|  | Normalized DeepSets | $\rho\left(1/n \sum_{i=1}^{n} \phi(\mathbf{x}_i)\right)$ | Bueno & Hylton (2021) |
|  | Hölder's Power DeepSets | $\rho\left(1/n \sum_{i=1}^{n} \phi(\mathbf{x}_i)^w\right)^{1/w}$ | Kimura et al. (2024) |
|  | QUANN-1 | $\rho\left(\psi^{-1}\left(1/n \sum_{i=1}^{n} \psi(\phi(\mathbf{x}_i))\right)\right)$ | (ours) |
| **Binary** | SetTransformer | $\rho\left(\sum_{(\mathbf{x}_i,\mathbf{x}_j)\in P_2(\mathbf{X})} \phi(\mathbf{x}_i, \mathbf{x}_j)\right)$ | Lee et al. (2019) |
|  | QUANN-2 | $\rho\left(\psi^{-1}\left(\sum_{(\mathbf{x}_i,\mathbf{x}_j)\in P_2(\mathbf{X})} \psi(\phi(\mathbf{x}_i, \mathbf{x}_j))\right)\right)$ | (ours) |

Any neural network that can be factorized in the above form, is said to be a $k$-ary Janossy pooling, distinct by its choice of $k$ and the adopted pooling operation (cf. Table 1). Below we discuss several important specific cases of Janossy pooling. It is important to note that alternative generalizations of the neural set functions were proposed (Kim et al., 2021).

**Unary pooling networks** Some of the earliest permutation-invariant deep learning architectures for set function modeling were DeepSets (Zaheer et al., 2017) and PointNet (Qi et al., 2017a). Both architectures can be viewed as special cases of unary Janossy pooling, where the permutation invariance is secured by using summation and max pooling, respectively.

**Binary pooling networks** Obviously, the unary pooling networks fail to capture the relationships between elements of the set, which imposes an important limitation. Inspired by the success of transformer architecture in various other application,Lee et al. (2019) proposed to addresses this limitation by introducing a SetTransformer, which takes into account the relationships between two elements in the input set via attention mechanism. However, SetTransformer can be factorized into a functional form in the equation 2 with $k = 2$ (Wagstaff et al., 2022), and thus constitutes a binary Janossy pooling.

**Janossy pooling with learnable pooling function** Clearly, the pooling function is the only non-learnable component in Janossy pooling (Eq. 2). This was first highlighted in Soelch et al. (2019) where the authors proposed using recurrent neural networks with attention as learnable alternative. However, the method's overall complexity, further aggravated by rather unclear implementation, limits its applicability to most real world datasets. Kimura et al. (2024) proposed Hölder's Power DeepSets (HPDS) that uses *power mean* with learnable exponent as the pooling function. Closely related to this work is the learnable aggregation function proposed by Pellegrini et al. (2021), in which four distinct power means are fused through a fixed-form aggregation scheme with additional learnable parameters. However, this method does not admit a strict Janossy factorization.

## 3.2 NON-JANOSSY METHODS

Not all permutation-invariant architectures can be factorized into the functional form described in Eq. 2. These non-Janossy approaches can be broadly grouped into two conceptually distinct categories: models that achieve permutation invariance through slot attention mechanisms and models that seek an optimal ordering of the input set.

**Slot attention** Slot attention models learn set representations by binding a fixed number of learnable "slots" to input set elements, using attention (Locatello et al., 2020). Each slot acts as a latent variable that competes to explain different aspects of the input through soft assignment and recurrent refinement. The input elements are transformed via linear projections and then aggregated through a weighted mean, where the attention coefficients serve as the weights. Multiple modification and extensions were later proposed (Skianis et al., 2020; Wu et al., 2022; Jia et al., 2023; Seitzer et al., 2023; Zhang et al., 2023).

**Permutation optimization and sorting** Zhang et al. (2019b) addressed permutation invariance by explicitly computing the optimal permutation of the input set with respect to a learnable cost function, thereby aligning elements before aggregation. In contrast, Zhang et al. (2020) proposed a simpler approach, where inputs are independently sorted along each feature dimension and subsequently fed to the encoder.

# 4 NEURALIZED KOLMOGOROV MEAN

In this section, we introduce *neuralized Kolmogorov means* (NKM) – a novel framework for aggregating sets of embeddings via a learnable measure of central tendency. In the following section we show how NKMs can be used as learnable pooling operation to improve set function approximation.

**Kolmogorov mean** The Kolmogorov mean, also known as the quasi-arithmetic mean, is a *measure of central tendency* that generalizes various common means. Given a continuous and invertible function $f$, the Kolmogorov mean of a set of numbers $\{x_i\}_{i=1}^n$ is defined as: $M_f = f^{-1}\big(1/n \sum_i^n f(x_i)\big)$, where the function $f$ is usually referred to as *generating function*. Since the generating function $f$ is continuous and invertible, it must be a strictly monotonic function on its domain, which guarantees that $M_f$ is bounded between $\min(x)$ and $\max(x)$. Special cases of Kolmogorov mean include, arithmetic ($f(x) = ax + b$), geometric ($f(x) = \log x$), and power means ($f(x) = x^p$).

**Neuralization of Kolmogorov mean** We propose a neuralized form of the Kolmogorov mean, where the generating function is implemented by an invertible neural network $\psi$:

$$M_\psi(\mathbf{X}) = \psi^{-1}\Big(\frac{1}{n}\sum_{i=1}^n \psi(\mathbf{x}_i)\Big) \tag{3}$$

This way we obtain learnable measure of central tendency, which we refer to as neuralized Kolmogorov mean (NKM). Despite the broad attention given by the scientific community (Kimura et al., 2024), to our knowledge, *we are the first to propose, not only neuralized, but any machine-learnable modification of the Kolmogorov mean*. Conceptually related, yet complementary, is the use of invertible neural functions for neuralization of semirings proposed by Dos Martires (2021).

**Choice of invertible architecture** In all our experiments, we used the RevNet (Gomez et al., 2017) architecture to implement the generating function $\psi$ of the NKM. Alternative choice of the invertible architecture in NKM were explored, but did not lead to a qualitative change in the models' performance (cf. Appendix 5).

# 5 QUASI-ARITHMETIC NEURAL NETWORKS

Intuitively, introducing learnable pooling operation to set function models should enhance the expressiveness of their hypothesis spaces. Therefore, we propose adopting the hereby introduced NKMs (cf. Section 4) as learnable pooling operation, resulting in a novel class of models, which we refer to as Quasi-Arithmetic Neural Networks (QUANNs). Note that we chose a distinct name to emphasize that NKM represents a general framework for learnable aggregation of latent codes, whereas QUANNs are specifically designed for set function learning.

## 5.1 RATIONALE

Any neural network that approximates a set function $F(\mathbf{X})$ using the following factorization is referred to as a k-ary QUANN, where the value of $k$ distinguishes the degree of interactions it models:

$$\hat{F}(\mathbf{X}) = \rho\Big[\psi^{-1}\Big(\frac{1}{|P_k(\mathbf{X})|} \sum_{\pi \in P_k(\mathbf{X})} \psi(\phi(\pi))\Big)\Big] \tag{4}$$

where $\phi, \rho$ are any arbitrary neural functions, while $\psi$ is invertible neural function to serve as generative function of the NKM. In more compact form, we can write $\rho(M_{\psi,\phi}(\mathbf{X}))$, where $M_{\psi,\phi}(\mathbf{X})$ indicates NKM induced via $\psi$ using encoder $\phi$.

Depending on the choice of $k$ we thus propose two specific instances of QUANN models: (i) unary pooling network QUANN-1 encoding individual set elements ($P_1(\mathbf{X}) = \mathbf{X}$); and (ii) binary pooling network QUANN-2 encoding interactions between the set element pairs via attention mechanism (similar to SetTransfomer)(cf. Table 1).

## 5.2 APPROXIMATION OF THE COMMON DECOMPOSITIONS

**Theorem 5.1** (Universal Approximation of QUANNs). *Let $\mathcal{U}$ denote the set of all permutation-invariant set functions $F : \mathcal{P}_f(X) \to \mathcal{Y}$ that can be uniformly approximated by Quasi-Arithmetic Neural Networks of the form 4, where $\rho$, $\psi$ and $\phi$ are arbitrary neural networks. Let $\mathcal{U}_W$ denote the set of all permutation-invariant set functions $F : \mathcal{P}_f(X) \to \dagger$ that are uniformly continuous with respect to the Wasserstein metric. Then $\mathcal{U} \supseteq \mathcal{U}_W$.*

**Means and central tendencies** QUANNs are fundamentally mean-pooling architectures, and as such, they are able to *exactly represent* any function that is *uniformly continuous with respect to the Wasserstein metric* (cf. Theorem 5.1). This means that any continuous function that depends smoothly on the empirical distribution of the input set, rather than single extreme point, can be approximated arbitrarily well using QUANNs. These involve various types of means and other central tendencies (e.g. median, quantiles).

This also means that QUANNs, in principle, are not able to exactly represent target set functions that are sum- or max-decomposable. However, in the context of set function learning, exact representation is seldom the objective; instead, the central goal is often to *approximate* a target function with sufficient accuracy (Wagstaff et al., 2022). To this end, we analyze QUANNs approximation capacity with respect to these decompositions.

**Max decomposition** Under mild assumptions, QUANNs can approximate max-decomposable set functions with worst-case approximation error that scales favorably with the cardinality of the input set (cf. Proposition H.4 – *Approximation of Max-Decomposable Set Function*). This is achieved by learning a generating function $\psi$ that falls into an exponential function family (cf. Table 2). In which case, for finite-size sets, QUANNs can attain arbitrarily small approximation error.

**Sum decomposition** QUANNs exhibit limited capacity in approximating sum-decomposable set functions due to the inherently expansive nature of summation, which poses a principal challenge for neural architectures designed to preserve permutation invariance via normalization-based pooling (Bueno & Hylton, 2021). Specifically, we demonstrate that, regardless of the particular function $\psi$ learned by the model, the approximation error tends to grow linearly with the cardinality $n$ of the input set, thus constraining the model's performance in sum-structured tasks (cf. Proposition H.5 – *Approximation of Sum-Decomposable Set Function*).

Table 2: Order of approximation $\mathcal{E}$ of mean, sum and max-decomposable set functions $F(x)$ that QUANNs can achieve by learning $\psi$ that falls under the given function family.

| $F(X)$ | $\psi$ | $\mathcal{E}$ |
|---|---|---|
| $a \circ \text{mean} \circ b$ | $w_1\mathbf{x} + w_2$ | $\mathcal{O}(1)$ |
| $a \circ \text{max} \circ b$ | $\exp(w\mathbf{x})$ | $\mathcal{O}(1/w \log n)$ |
| $a \circ \text{sum} \circ b$ | $\cdot$ | $\mathcal{O}(n)$ |

However, QUANNs can still approximate such functions so that *the expected approximation error becomes arbitrarily small* – by letting the encoder $\phi$ "absorb" the expected value of the inputs cardinality and thus to appropriately rescale the learned NKM (cf. Proposition H.6 – *Expected value of the Approximation of Sum-Decomposable Set Function*).

## 5.3 THEORETICAL BENEFITS OVER NORMALIZED DEEPSETS

A careful reader may notice a formal similarity between the functional form of the normalized DeepSet and that of the proposed QUANN model. Specifically, by introducing substitutions $\rho' = \rho \circ \psi^{-1}$ and $\phi' = \psi \circ \phi$, the functional form of QUANNs can be rewritten in a form that is functionally equivalent to normalized DeepSets: $\rho'(1/n \sum_{i=1}^{n} \phi'(\mathbf{x}_i))$. This may suggest that QUANNs and Normalized DeepSets are essentially the same model class. However, in such case, $\rho'$ and $\phi'$ would not be independent functions, but functional compositions connected via component $\psi$ and its inverse. This distinction confers several important advantages to the QUANN architecture.

**Increased expressivity under fixed $\rho$ and $\phi$**   First, under any fixed choice of the encoder $\phi$ and the estimator $\rho$ in a normalized DeepSet, introducing the additional learnable and invertible transformation $\psi$ strictly increases the expressivity of the model. This expansion of the hypothesis space can be particularly valuable in practical scenarios where one or more components (e.g., the encoder $\phi$) are pre-trained, such as in transfer learning.

**Natural regularization via $\psi$**   Enforcing invertibility on $\psi$ acts as a natural form of regularization. We hypothesize that this constraint improves the generalization capability of the model by encouraging the network to learn a $\psi$ that aligns well with the intrinsic structure or decomposition of the target set function $F(\mathbf{X})$; independent of the specific choices of the encoder or estimator networks.

**Dimensional collapse prevention**   To better illustrate this point, consider the derivative of the function $\hat{F}(\mathbf{X})$ with respect of the $w_\phi$, the parameter of the encoder $\phi$. For simplicity, we assume that $\hat{F}(\mathbf{X})$ is factorized as QUANN-1 model. Using the *inverse function theorem* together with the derivative of Kolmogorov mean (cf. Appendix A), we obtain:

$$\frac{\partial \hat{F}(\mathbf{X})}{\partial w_\phi} = \mathbf{J}_\rho(M_{\psi,\phi}) \mathbf{J}_\psi^{-1}(M_{\psi,\phi}) \frac{1}{|\mathbf{X}|} \sum_{\mathbf{x} \in \mathbf{X}} \mathbf{J}_\psi(\phi(\mathbf{x})) \frac{\partial \phi(\mathbf{x})}{\partial w_\phi} \tag{5}$$

$$= \mathbf{J}_\rho(M_{\psi,\phi}) \sum_{\mathbf{x} \in \mathbf{X}} \frac{1}{|\mathbf{X}|} \mathbf{J}_\psi^{-1}(M_{\psi,\phi}) \mathbf{J}_\psi(\phi(\mathbf{x})) \frac{\partial \phi(\mathbf{x})}{\partial w_\phi} \tag{6}$$

$$= \mathbf{J}_\rho(M_{\psi,\phi}) \sum_{\mathbf{x} \in \mathbf{X}} \frac{\partial M_{\psi,\phi}}{\partial \phi(\mathbf{x})} \frac{\partial \phi(\mathbf{x})}{\partial w_\phi} \tag{7}$$

where $\mathbf{J}_\rho$ denotes the Jacobian of the estimator $\rho$. During training, the derivative of Kolmogorov mean $\partial M_{\psi,\phi}/\partial \phi(\pi)$ scales changes in the latent representations of the elements $\mathbf{x}$ proportionally to their "leverage" on the representation of the entire input set – $M_{\psi,\phi}$. Because $\psi$ is invertible, derivative of Kolmogorov mean is *non-singular*, ensuring that no latent dimension is collapsed to zero when aggregating gradients across the inputs. Consequently, information from each dimension is preserved during backpropagation, effectively *preventing dimensional collapse* (Jing et al., 2022).

**Structured latent representations**   If $\psi$ is learned to be locally monotonic around $M_{\psi,\phi}$, then the derivative $\partial M_{\psi,\phi}/\partial \phi(\pi)$ in Equation 7 is *positive semidefinite*. In this case, $M_{\psi,\phi}$ tends to *follow* the gradients computed across the inputs and stays within a *ball* around the latent representations of the input set elements (Nielsen, 2023), leading to more structured latent representations.

## 6   EXPERIMENTAL DESIGN

**Research questions**   We hypothesize that the theoretical advantages established in the previous section enable QUANNs to learn more transferable encoders and achieve superior performance compared to state-of-the-art methods. Based on this hypothesis, we formulate the following research questions. **RQ1:** Does the proposed neural function factorization described by Equation 4 improve the approximation of common set function decompositions compared to an equivalent model without the use of invertible neural networks? **RQ2:** Does the proposed approach learn a better encoder $\phi$, by (a) producing qualitatively improved embeddings compared to those trained using baseline methods, (b) reducing the need for fine-tuning pre-trained encoders in downstream set function learning tasks, and (c) enabling the learned encoders to be transferable to non-set tasks? **RQ3:** Do QUANNs outperform state-of-the-art methods in set function learning tasks across diverse experimental conditions?

**Baselines**   To evaluate the performance of our proposed method, we compared it against a diverse set of Janossy pooling models. We included three unary pooling models: **DeepSets** (Zaheer et al., 2017) and **PointNet** (Qi et al., 2017a), which are widely regarded as foundational methods, and **Normalized DeepSets** (Bueno & Hylton, 2021), which serves as a representative method for learning average decomposable functions. As a representative of Janossy pooling methods with a learnable pooling function, we employed the **Hölder Power Deep Set (HPDS)** model (Kimura et al., 2024). For binary pooling architectures, we used the **SetTransformer** (Lee et al., 2019), which conveys

set function approximation via pairwise attention-based interactions between set elements. In addition, we compared QUANNs with three representatives of non-Janossy methods: **Slot attention (SlotAtt)** (Locatello et al., 2020), **FSPool** (Zhang et al., 2020) and **LAF** (Pellegrini et al., 2021).

**Datasets** To evaluate our method against baseline models, we adopted one **synthetic dataset** and three publicly available real-world datasets: **MNIST** (LeCun et al., 1998), **Omniglot** (Lake et al., 2015) and **ModelNet40** (Wu et al., 2015). Since MNIST and Omniglot, are not originally designed for set function modeling, similar to previous works (Zaheer et al., 2017; Lee et al., 2019), we introduced modifications to adapt these datasets to the set-based setting. The specific preprocessing procedures and task definitions are detailed in the Section E.1.

## 7 RESULTS

### 7.1 SYNTHETIC DATA EXPERIMENTS

We designed a synthetic data experiment using randomly generated point clouds and 10 different vector aggregation functions, which models were supposed to approximate (cf. Section E.1). We evaluated QUANN-1 against three models: Ablation 1, in which the transformation $\psi$ was replaced with the identity function: $\psi(\mathbf{x}) = \mathbf{x}$, Ablation 2, which also replaced $\psi$ with the identity function, but extended the encoder $\phi$ and the estimator $\rho$ networks such that the total number of trainable parameters and the model depth approximately matched those of QUANN-1; and Ablation 3, which removed $1/|P_k(\mathbf{X})|$ normalization in the Equation 4. QUANN-1 outperformed the ablated models in 9 out of 10 experiments (RQ1).

Table 3: Results of the synthetic data experiments. Performance of the proposed model (QUANN-1) and the three ablated models when approximating 10 different set aggregation functions, as measured by mean squared error (MSE, lower is better). QUANN-1 outperformed ablations in 9 out of 10 experiments (highlighted in bold); with significant improvements in 7 of them.

| | Ablation 1 | | Ablation 2 | | Ablation 3 | | QUANN-1 | |
|---|---|---|---|---|---|---|---|---|
| | Mean | Std | Mean | Std | Mean | Std | Mean | Std |
| Geometric median | 4.521 | 0.213 | 14.955 | 0.704 | 14.096 | 1.228 | **4.138** | 0.453 |
| Quadratic mean | 1.011 | 0.122 | 3.643 | 0.101 | 6.667 | 0.805 | **0.988** | 0.111 |
| Median | 4.628 | 0.135 | 15.236 | 0.781 | 14.171 | 1.028 | **4.104** | 0.383 |
| Medoid | 18.725 | 0.723 | 31.075 | 1.243 | 27.179 | 1.337 | **17.312** | 0.667 |
| Midpoint | 25.914 | 0.571 | 38.691 | 0.915 | 33.578 | 0.965 | **24.236** | 0.497 |
| VecMaxNorm | 94.926 | 2.440 | 137.045 | 4.350 | 114.042 | 1.933 | **89.499** | 2.363 |
| Max | 20.929 | 0.817 | 43.143 | 1.854 | 45.038 | 6.135 | **13.583** | 0.565 |
| LogSumExp | 19.242 | 1.078 | 41.386 | 1.463 | 43.969 | 7.656 | **12.557** | 0.856 |
| Variance | 132.388 | 10.578 | 105.119 | 9.594 | 589.282 | 95.461 | **74.128** | 15.605 |
| Skewness | 0.027 | 0.001 | 0.027 | 0.001 | 0.074 | 0.032 | 0.027 | 0.001 |

### 7.2 MNIST-SETS EXPERIMENTS

We expanded the sum-of-MNIST-digits experiment, originally performed in Zaheer et al. (2017), by adding other aggregating functions, including max, mean, mode, etc. The model task is to learn to approximate the MNIST digits aggregation function $F$ by learning to estimate the function outputs (cf. Figure 2). The experiments were performed under two experimental setups: using generic encoders, which was followed by the qualitative analysis; and using pre-trained encoders (neural function $\phi$).

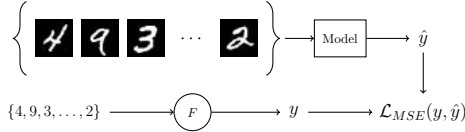

Figure 2: Experiment with aggregation of MNIST digits. The goal is to approximate the function $F$ by learning to estimate its outputs.

**Using generic encoders** The MNIST-set experiments were first conducted using generic encoders, where the models had to learn the MNIST image embeddings from scratch alongside other neural components. According to the obtained results (cf. Table 5), QUANN-1 outperformed all unary baselines across all 5 aggregation tasks. The second proposed model, QUANN-2, achieved overall the best performance across all models evaluated. These experiments were further expanded by

adding five measures of central tendency; providing further validation of the benefits of QUANN models (cf. Supplementary Table F).

**Qualitative analysis**    To assess the models' ability to learn meaningful input representations, we extracted the image encoders $\phi$ from the Janossy models trained in the previous experiments and applied them to embed the MNIST test set images. The resulting embeddings were subsequently projected into a two-dimensional space using UMAP (McInnes et al., 2018). The obtained projections were then plotted to reveal the structure of the embeddings (cf. Figure 7). The results show that only the embeddings obtained from QUANNs, and SetTransformer, but not other models, consistently produced class-wise separable structures across all tasks, demonstrating their ability to learn more structured representations (RQ2a).

**Using pre-trained encoders**    The experiments were then repeated using pre-trained MNIST image encoders (neural function $\phi$). Specifically, we first trained a standalone MNIST image classifier composed of a small convolutional neural network followed by an MLP predictor, achieving 98% test set accuracy. The trained encoder was transferred to the set function models, with its weights being fixed. The models were then trained in the same manner as in the previous experiments.

Since the encoder is trained to produce class-separated embeddings and not to recover quantitative relations between the images, we expected that the performance of the set function models would generally worsen under this setting. The goal was to assess whether QUANNs would experience a smaller performance drop compared to the baselines. This was confirmed by the results, which show that only QUANNs retain their performance across all tasks, whereas the baselines experience significant performance drop (cf. Figure 8); despite similar number of trainable parameters (RQ2b).

## 7.3    OMNIGLOT-SETS EXPERIMENTS

We modified the experiment proposed in Lee et al. (2019) to a multi-label binary classification, where the goal was to identify which of the 40 Omniglot alphabets were represented within a given set of $n$ Omniglot handwritten character images (cf. Figure 3). The experiments were conducted under four different conditions, fixing the minimum cardinality at $n_{\min} = 5$ while varying the maximum cardinality of the image sets as $n_{\max} = 10, 15, 20$ and $25$.

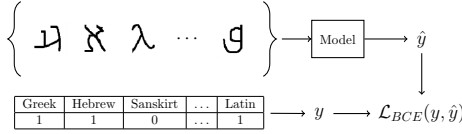

Figure 3: Omniglot experiment. The task is to identify which alphabets are represented in a set of images of hand-written characters.

The performance was assessed using the balanced accuracy score, computed as the macro-average across the alphabets. The obtained results (cf. Table 5) show that QUANNs surpassed all baselines.

**Transfer learning**    We extended the Omniglot experiments with a transfer learning analysis. We extracted the encoders $\phi$ from all models trained in the Omniglot experiment, froze their weights, and transferred them to a image classification model composed of the encoder followed by a linear layer. This model was trained to predict the alphabet to which each image belongs.

The results show that encoders learned by QUANNs provide better support for the image classification compared to those learned by the baselines (cf. Table 4). This suggests that the factorization employed in QUANNs learns more transferrable embeddings that can be used beyond the set-related tasks (RQ2a).

Table 4: Omniglot image classification obtained from the transfer learning experiment.

|  |  | ACC ↑ | |
| --- | --- | --- | --- |
|  |  | Mean | Std |
|  | Model | | |
| Unary | DeepSet | 0.524 | 0.042 |
|  | PointNet | 0.256 | 0.219 |
|  | NormDeepSet | 0.393 | 0.052 |
|  | HPDS | 0.435 | 0.023 |
|  | QUANN-1 | **0.597** | 0.014 |
| Binary | SetTransformer | 0.480 | 0.019 |
|  | QUANN-2 | 0.589 | 0.041 |
| Non-Janossy | FSPool | 0.463 | 0.051 |
|  | SlotAtt | 0.484 | 0.018 |
|  | LAF | 0.156 | 0.007 |

## 7.4    MODELNET40 EXPERIMENTS

We performed multi-class classification experiments using the ModelNet40 dataset, following a setup similar to that of Qi et al. (2017a) and Qi et al. (2017b). The task was to predict the category

of the object represented by a point cloud. The experiments were conducted under three different settings: the point clouds were subsampled to $n$ points, where the maximum number of points was fixed at $n_{\max} = 2048$, while the minimum number of points varied as $n_{\min} = 256, 512$ and $1024$.

QUANNs were outperformed by FSPool and also by PointNet, but not other baselines (cf. Table 5). The high performance of FSPool and PointNet is not surprising, as both models employ architectures that are well-suited for detecting extreme values (max or min), a property known to be critical for capturing local and global geometric features of point clouds (Zhang et al., 2019a). Overall, these results corroborate the findings reported in the original experiments (Qi et al., 2017a).

## 7.5 QM9 EXPERIMENTS

Finally, we evaluated our models on two regression tasks derived from the QM9 molecular dataset (Ramakrishnan et al., 2014). Each molecule is represented as a set of atoms with associated 3D coordinates, ranging from 3 to 29 atoms per molecule. In the first task, the goal was to predict the HOMO energy level, while in the second task we predicted the LUMO energy level. As shown in Table 5, QUANN-2 achieves the best performance across all baselines on both tasks.

## 8 DISCUSSION

**Conclusions** In this work, we introduced QUANNs, a novel approach for set function approximation that leverages a neuralized Kolmogorov mean as a trainable pooling operation. We presented theoretical benefits of this construction and hypothesized that these advantages should promote the learning of more transferable latent representations and improved generalization. The obtained empirical results provide evidence in support of these hypotheses and show that QUANNs significantly outperform their state-of-the-art alternatives (cf. Figure 4) (RQ3).

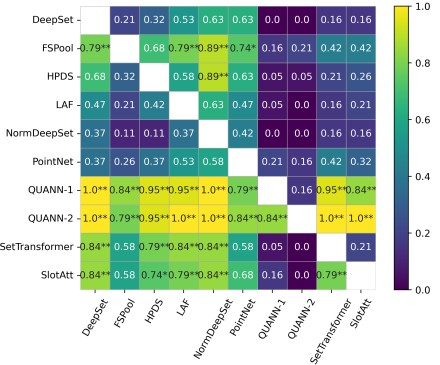

Figure 4: Relative number of outcomes (cf. Table 5), where the model in the given row outperformed the model in the given column. An asterisk and a double asterisk indicate statistical significance ($p < 0.05$) and very high significance ($p < 0.01$), respectively.

**Limitations** As the major limitation of QUANNs we consider their limited capacity in approximating sum-decomposable set functions (cf. Section 5.2). Therefore, if the target set function is expected to be additive or expansive, a simple sum may provide better alternative to NKM. Similarly, when data are limited, NKM may be unnecessarily flexible and prone to overfitting, whereas a fixed function may provide better inductive bias and computational simplicity.

Similar to most previous works, our theoretical and experimental results are restricted to countable sets. Intuitively, extending QUANNs to uncountable sets may be possible by replacing the sum in Eq. 4 with an appropriate form of computational integration. However, we cannot guarantee which of the theoretical benefits of our method would carry over in this setting, and the approximation of set functions over uncountable domains remains an open question.

**Applications** QUANNs provide a general and expressive framework for learning central tendencies over arbitrary collections of representations, making them potentially useful across a variety of domains. For instance, QUANNs could be employed within graph neural networks, replacing conventional message-passing aggregators with a learnable generalized mean, which is fundamentally different from the approaches of Corso et al. (2020) and Ong & Veličković (2022), who use learnable fusion of simple aggregates and learnable pairwise operations, respectively. Similarly, in multi-modal and multi-view learning (Guo et al., 2019), where information from several modalities or sources must be fused, QUANNs offer a principled way to learn the fusion rule for combining representations from each view. QUANNs could also be applied in federated learning (Guendouzi et al., 2023), where model updates from multiple clients must be aggregated.

Table 5: Model performance across all experiments, grouped by model class (Unary in the upper table; Binary and Non-Janossy in the lower). All values report mean performance across experimental replicates with corresponding standard deviations. Bold and underlined values denote the best and second-best performance across all models, respectively.

(a) *Unary models*

| | DeepSet | | PointNet | | NormDeepSet | | HPDS | | QUANN-1 | |
| | Mean | Std | Mean | Std | Mean | Std | Mean | Std | Mean | Std |
|---|---|---|---|---|---|---|---|---|---|---|
| Agg. | | | | | *MNIST, MSE ↓* | | | | | |
| max | 0.120 | 0.012 | 0.079 | 0.002 | 0.128 | 0.014 | 0.071 | 0.009 | **0.067** | 0.022 |
| mean | 0.095 | 0.013 | 0.115 | 0.011 | 0.089 | 0.005 | 0.047 | 0.005 | 0.023 | 0.004 |
| mode | 3.903 | 0.150 | 5.498 | 0.270 | 4.119 | 0.170 | 4.011 | 0.295 | 2.219 | 0.163 |
| sum | 5.924 | 0.694 | 18.740 | 0.921 | 7.706 | 1.575 | 5.844 | 0.738 | 4.066 | 0.673 |
| variance | 1.747 | 0.226 | 2.143 | 0.121 | 1.927 | 0.132 | 1.265 | 0.414 | 0.605 | 0.171 |
| Agg. | | | | | *MNIST - Pre-trained encoders, MSE ↓* | | | | | |
| max | 0.421 | 0.041 | 0.199 | 0.018 | 0.408 | 0.025 | 0.314 | 0.068 | 0.045 | 0.014 |
| mean | 0.207 | 0.008 | 0.317 | 0.023 | 0.201 | 0.005 | 0.153 | 0.006 | 0.021 | 0.003 |
| mode | 3.880 | 0.143 | 6.384 | 0.170 | 3.626 | 0.095 | 3.300 | 0.085 | 1.043 | 0.337 |
| sum | 15.241 | 1.129 | 83.217 | 2.177 | 17.281 | 1.756 | 18.115 | 2.764 | 3.074 | 0.313 |
| variance | 4.245 | 0.086 | 3.766 | 0.182 | 3.950 | 0.173 | 3.774 | 0.296 | 0.625 | 0.212 |
| $n_{max}$ | | | | | *Omniglot, Balanced ACC ↑* | | | | | |
| 10 | 0.579 | 0.009 | 0.566 | 0.076 | 0.544 | 0.011 | 0.610 | 0.017 | **0.734** | 0.007 |
| 15 | 0.584 | 0.007 | 0.523 | 0.036 | 0.519 | 0.001 | 0.535 | 0.015 | 0.711 | 0.017 |
| 20 | 0.588 | 0.007 | 0.515 | 0.012 | 0.527 | 0.002 | 0.543 | 0.011 | 0.674 | 0.033 |
| 25 | 0.592 | 0.004 | 0.545 | 0.077 | 0.536 | 0.006 | 0.537 | 0.009 | **0.655** | 0.001 |
| $n_{min}$ | | | | | *ModelNet40, ACC ↑* | | | | | |
| 256 | 0.648 | 0.007 | 0.788 | 0.005 | 0.649 | 0.006 | 0.658 | 0.009 | 0.676 | 0.003 |
| 512 | 0.649 | 0.005 | 0.788 | 0.002 | 0.520 | 0.011 | 0.667 | 0.007 | 0.688 | 0.013 |
| 1024 | 0.645 | 0.012 | 0.791 | 0.006 | 0.534 | 0.007 | 0.663 | 0.013 | 0.686 | 0.013 |
| Task | | | | | *QM9, MSE ↓* | | | | | |
| homo | 179.423 | 10.715 | 153.419 | 7.299 | 166.695 | 6.495 | 212.237 | 7.899 | 160.345 | 12.676 |
| lumo | 399.119 | 13.734 | 401.247 | 3.917 | 458.711 | 10.189 | 674.320 | 20.098 | 374.646 | 9.096 |

(b) *Binary & Non-Janossy models*

| | Binary | | | | Non-Janossy | | | | | |
| | SetTransformer | | QUANN-2 | | FSPool | | SlotAtt | | LAF | |
| | Mean | Std | Mean | Std | Mean | Std | Mean | Std | Mean | Std |
|---|---|---|---|---|---|---|---|---|---|---|
| Agg. | | | | | *MNIST, MSE ↓* | | | | | |
| max | 0.099 | 0.009 | 0.069 | 0.014 | 0.095 | 0.018 | 0.085 | 0.025 | 0.571 | 0.092 |
| mean | 0.031 | 0.002 | **0.023** | 0.003 | 0.075 | 0.007 | 0.029 | 0.005 | 0.050 | 0.012 |
| mode | 3.085 | 0.367 | **1.579** | 0.408 | 2.847 | 0.479 | 3.255 | 0.124 | 4.219 | 0.370 |
| sum | 4.090 | 0.601 | **2.861** | 0.251 | 5.202 | 0.441 | 66.324 | 5.957 | 20.920 | 3.681 |
| variance | 2.284 | 0.298 | **0.605** | 0.043 | 1.317 | 0.201 | 1.956 | 0.120 | 1.383 | 0.197 |
| Agg. | | | | | *MNIST - Pre-trained encoders, MSE ↓* | | | | | |
| max | 0.079 | 0.016 | **0.042** | 0.008 | 0.230 | 0.038 | 0.077 | 0.013 | 0.653 | 0.045 |
| mean | 0.046 | 0.006 | **0.020** | 0.002 | 0.142 | 0.019 | 0.049 | 0.002 | 0.101 | 0.022 |
| mode | 2.837 | 0.185 | **0.756** | 0.169 | 3.475 | 0.381 | 2.556 | 0.178 | 2.851 | 0.911 |
| sum | 4.725 | 0.456 | **2.535** | 0.377 | 9.520 | 0.509 | 61.432 | 5.679 | 14.319 | 5.439 |
| variance | 2.322 | 0.351 | **0.616** | 0.134 | 2.583 | 0.241 | 1.825 | 0.026 | 1.775 | 0.321 |
| $n_{max}$ | | | | | *Omniglot, Balanced ACC ↑* | | | | | |
| 10 | 0.525 | 0.050 | 0.723 | 0.011 | 0.557 | 0.074 | 0.613 | 0.006 | 0.508 | 0.008 |
| 15 | 0.603 | 0.003 | **0.711** | 0.015 | 0.579 | 0.079 | 0.618 | 0.005 | 0.528 | 0.003 |
| 20 | 0.606 | 0.004 | **0.693** | 0.017 | 0.575 | 0.060 | 0.618 | 0.009 | 0.535 | 0.006 |
| 25 | 0.606 | 0.006 | 0.635 | 0.069 | 0.640 | 0.012 | 0.616 | 0.005 | 0.545 | 0.010 |
| $n_{min}$ | | | | | *ModelNet40, ACC ↑* | | | | | |
| 256 | 0.676 | 0.006 | 0.773 | 0.011 | **0.796** | 0.011 | 0.726 | 0.003 | 0.660 | 0.020 |
| 512 | 0.679 | 0.009 | 0.762 | 0.011 | **0.791** | 0.013 | 0.739 | 0.014 | 0.671 | 0.028 |
| 1024 | 0.678 | 0.009 | 0.782 | 0.005 | **0.794** | 0.012 | 0.727 | 0.012 | 0.688 | 0.008 |
| Task | | | | | *QM9, MSE ↓* | | | | | |
| homo | 165.998 | 2.392 | **104.821** | 10.472 | 176.240 | 7.769 | 164.640 | 19.453 | 297.712 | 84.158 |
| lumo | 461.948 | 11.837 | **258.039** | 20.471 | 480.726 | 32.999 | 386.630 | 11.782 | 657.083 | 77.112 |

ACKNOWLEDGEMENTS

This work was supported by the Institute of Information & Communications Technology Planning & Evaluation (IITP) grant funded by the Korean Government (MSIT) (No. RS-2024-00457882, National AI Research Lab Project).

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

## A  DERIVATIVE OF THE KOLMOGOROV MEAN

We analyze the derivative of the Kolmogorov mean with respect to a single input element $x_j$. Given an invertible and differentiable transformation $\psi$, the Kolmogorov mean of a set $\mathbf{X} = \{\mathbf{x}_1, \ldots, \mathbf{x}_n\}$ is defined as:

$$M_f(\mathbf{X}) = f^{-1}\left(\frac{1}{n}\sum_{i=1}^{n} f(\mathbf{x}_i)\right). \tag{8}$$

Let $\mathbf{z}_i = f(\mathbf{x}_i)$ and define the aggregated representation $\bar{\mathbf{z}} = \frac{1}{n}\sum_{i=1}^{n} \mathbf{z}_i$. Then:

$$\mathbf{y} = M_f(\mathbf{X}) = f^{-1}(\bar{\mathbf{z}}).$$

Our goal is to compute the *sensitivity* of $\mathbf{y}$ with respect to a single input $\mathbf{x}_j$. Applying the multivariate chain rule:

$$\frac{\partial M_f(\mathbf{X})}{\partial \mathbf{x}_j} = \frac{\partial f^{-1}(\bar{\mathbf{z}})}{\partial \bar{\mathbf{z}}} \cdot \frac{\partial \bar{\mathbf{z}}}{\partial \mathbf{x}_j}. \tag{9}$$

The second term is straightforward:

$$\frac{\partial \bar{\mathbf{z}}}{\partial \mathbf{x}_j} = \frac{1}{n}\frac{\partial \mathbf{z}_j}{\partial \mathbf{x}_j} = \frac{1}{n}J_f(\mathbf{x}_j), \tag{10}$$

where $J_f(\mathbf{x}_j) = \frac{\partial f(\mathbf{x}_j)}{\partial \mathbf{x}_j}$ is the Jacobian of $f$ at $\mathbf{x}_j$.

For the first term, we use the *inverse function theorem*: the Jacobian of the inverse map is the inverse of the Jacobian of the forward map:

$$\frac{\partial f^{-1}(\bar{\mathbf{z}})}{\partial \bar{\mathbf{z}}} = J_{f^{-1}}(\bar{\mathbf{z}}) = [J_f(\mathbf{y})]^{-1}, \quad \text{where } \mathbf{y} = M_f(\mathbf{X}). \tag{11}$$

Combining these results, we obtain:

$$\frac{\partial M_f(\mathbf{X})}{\partial \mathbf{x}_j} = \frac{1}{n}[J_f(\mathbf{y})]^{-1} \cdot J_f(\mathbf{x}_j) \tag{12}$$

**Key properties**  Since $f$ is invertible by the definition of Kolmogorov mean, its Jacobian $J_f$ is non-singular. Therefore, the resulting derivative $\partial M_f(\mathbf{X})/\partial \mathbf{x}_j$ is also guaranteed to be non-singular. Moreover, if $f$ is also monotonic, the resulting derivative is positive (in the scalar case) or has a positive-definite symmetric part (in the multivariate case). The resulting derivative of the Kolmogorov mean thus inherits at least two important structural properties:

- **Dimension preservation** (from non-singularity) – NKM maps any infinitesimal input perturbation $\delta \mathbf{x}_j$ to an output perturbation $\delta \mathbf{y}$ so that $\delta \mathbf{y}$ is non-zero along all non-zero dimensions of $\delta \mathbf{x}_j$. The NKM thus acts as a smooth aggregator that maps small input neighborhoods to small latent neighborhoods so that no dimension is flattened to zero.

- **Local alignment** (from positive-definiteness) – NKM maps any infinitesimal input perturbation $\delta \mathbf{x}_j$ to an output perturbation $\delta \mathbf{y}$ that satisfies $\delta \mathbf{y}^\top \delta \mathbf{x}_j > 0$ ensuring that small input perturbations produce output changes locally aligned with the input shift.

## B  CHOICE OF $\psi$-NETWORK ARCHITECTURE

In all our experiments, we used the RevNet (Gomez et al., 2017) architecture to implement the generating function $\psi$ of the NKM. RevNet is composed of a sequence of invertible blocks, where each block operates on a partitioned input vector $(x_1, x_2)$ and is defined as:

$$\begin{aligned} x_1' &= x_1 + f(x_2), \\ x_2' &= x_2 + g(x_1'), \end{aligned} \tag{13}$$

where $f$ and $g$ are arbitrary neural networks (we used ReLU-activated MLPs), and the transformation is easily invertible by sequentially subtracting these updates in reverse. We selected RevNet for its simplicity and efficiency, making it an intuitive choice for learning $\psi$.

To further investigate the relationship between the performance of QUANN and the architectural design of the neural network implementing the generating function $\psi$, we repeated the synthetic data experiments (cf. Section 7.1) using only QUANN models, under different architectures implementing the generating function $\psi$.

**Use of RealNVP coupling** The RevNet blocks used to implement the generating function $\psi$ were replaced with conceptually similar but more expressive RealNVP coupling blocks (Dinh et al., 2014; 2016). This modification preserved the invertibility constraint while increasing the representational capacity of $\psi$. The results (cf. Figure 5) did not reveal clear consistent pattern indicating improvement or worsening of the model performance, indicating that while architectural enhancements may yield marginal gains in some tasks, the overall effectiveness of QUANNs does not critically depend on the specific choice of invertible block, provided it is sufficiently expressive and scalable.

**Varying RevNet capacity** We systematically varied the number of RevNet blocks and the number of hidden layers per block. The resulting performance metrics, summarized in Figure 6, indicate that the impact of these architectural choices is highly task-dependent. That is, no single configuration consistently outperforms others across all aggregation functions. These findings suggest that, in practice, it is not possible to determine the optimal design of $\psi$ *a priori*. Instead, users are advised to empirically explore different configurations in order to identify the most effective architecture for a given task.

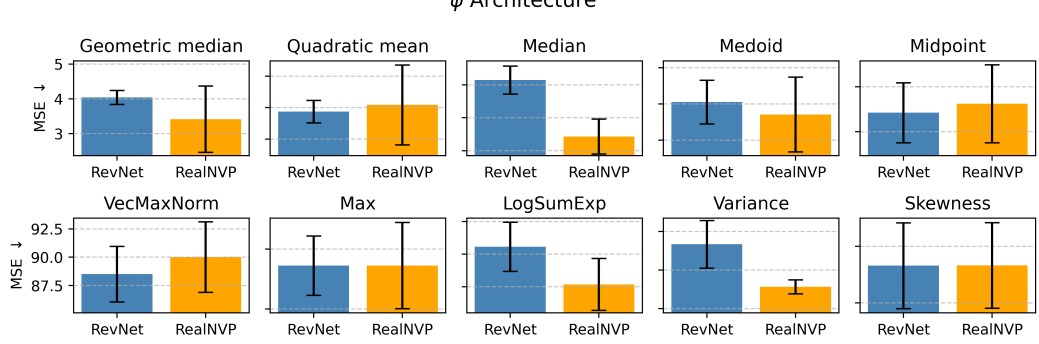

Figure 5: QUANN performance in synthetic data experiment in dependence on the choice of the $\psi$-network architecture (RevNet vs RealNVP coupling). The results show that there is no clear consistent pattern indicating improvement or worsening of the model performance due to altered network architecture.

## C  SCALABILITY OF NEURALIZED KOLMOGOROV MEANS

The NKM computation scales linearly with the number of elements in the input set: $\mathcal{O}(n)$, involving $n$ forward passes through the generating function $\psi$, followed by a single pass through its inverse $\psi^{-1}$. The scaling behavior with respect to the input dimensionality, $d$, depends on the choice of neural architecture used to implement $\psi$. When $\psi$ is realized as a RevNet, the memory and computational costs scale linearly with $d$: $\mathcal{O}(d)$. Overall, these properties make NKM a practical and scalable choice for processing even large and high-dimensional input sets.

## D  POSSIBLE EXTENSIONS & MODIFICATIONS OF QUANNS

In our experiments, we purposefully refrained from using regularization techniques or adopting more complex architectures, to maintain a clear epistemological focus on the core properties of the proposed models. Nevertheless, we identify at least three promising avenues for extension and

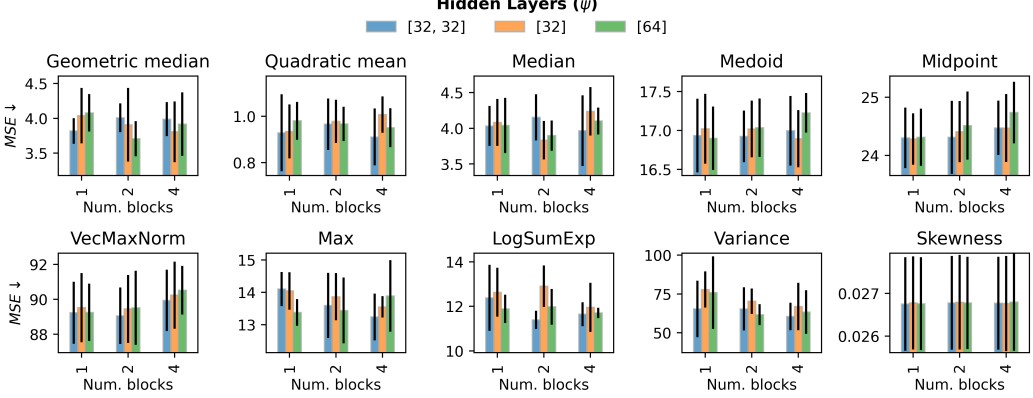

Figure 6: QUANN performance in synthetic data experiment in dependence on the choice of the $\psi$-network architecture (number of RevNet blocks and hidden layers per block). The results show that there is not universal relationship between the choice of the architecture and the associated performance.

modification of our work: (i) the adoption of regularization techniques, particularly set-specific normalization layers such as SetNorm (Zhang et al., 2022); (ii) hierarchical extensions of QUANNs, analogous to how PointNet++ (Qi et al., 2017b) extends the original PointNet (Qi et al., 2017a) and (iii) the introduction of *"multi-head" learnable pooling*, where multiple Neuralized Kolmogorov Means (with independently trained generating functions) are applied in parallel and then concatenated, may substantially boost expressivity of the models.

### D.1 Permutation-equivariant QUANNs

Our proposed framework focuses on learning permutation-invariant functions. A common strategy to extend permutation-invariant models to permutation-equivariant ones is to augment each element with global, sum-pooled features and then apply an element-wise function to combine local and global information (*e.g.*, (Segol & Lipman, 2020)). Analogously, permutation-equivariant QUANNs layer can be obtained by augmenting elements after applying the inverse transformation:

$$f_i(\mathbf{X}) = \rho\left(\mathbf{x}_i, \psi^{-1}\frac{1}{|P_k(\mathbf{X})|}\sum_{\pi \in P_k(\mathbf{X})}\psi\big(\phi(\mathbf{x}_i)\big)\right) \tag{14}$$

To evaluate the performance of permutation-equivariant layers, we conducted the following experiment. We implemented the permutation-equivariant variant of DeepSets as described in Zaheer et al. (2017) (Section 3.1, Equivariant Model, and Appendix C). We then assessed this model under different pooling operations using the ModelNet40 dataset, strictly adhering to the preprocessing protocol outlined in Zaheer et al. (2017) (Section H), including zero-mean/unit-variance normalization and fixed set cardinality ($n = 100$ and $n = 1000$). The best performance we obtained closely matched the results originally reported in Zaheer et al. (2017).

Table 6: ModelNet40 classification obtained by the permutation-equivariant models.

| Model | ACC↑ | | | |
|---|---|---|---|---|
| | $n = 100$ | | $n = 1000$ | |
| | Mean | Std | Mean | Std |
| DeepSet | 0.798 | 0.021 | 0.844 | 0.017 |
| **QUANN-1** | 0.786 | 0.015 | 0.835 | 0.022 |
| **QUANN-2** | **0.846** | 0.012 | **0.882** | 0.018 |

We then implemented an analogous model in which the equivariant DeepSets layers were replaced with QUANN-based equivariant layers as defined in Equation 14. While the equivariant QUANN-1 achieved performance comparable to that of equivariant DeepSets, the equivariant QUANN-2 model substantially outperformed both, as shown in Table 6.

# E  EXPERIMENTS DETAILS

## E.1  DATASETS

**Synthetic dataset**  Each point cloud $X = (x_i)_{i=1}^n$ consisted of $n \in \mathbb{N}$ vectors randomly sampled from $\mathbf{x} \in (0,1)^{16}$. Number of vectors $n$ was sampled uniformly from interval $[2, 1024]$. The obtained vectors were subsequently transformed using affine transformations of the form $ax + b$, where $a \in (0,1)$ is a scalar and $b \in (-10, 10)^{16}$ is a translation vector sampled independently for each cloud. To assign labels, we applied a selected aggregation function $F(\mathbf{X}) : \mathbb{R}^d \to \mathbb{R}^d$, computed across the set of transformed points. We experimented with a total of ten distinct aggregation functions, encompassing a range of statistical descriptors (cf. Table 8). The resulting dataset was partitioned into training, validation, and test subsets consisting of $20 \times 10^3$, $2 \times 10^3$, and $3 \times 10^3$ clouds respectively.

**MNIST-sets dataset**  MNIST is a widely used benchmark dataset consisting of grayscale $28 \times 28$ pixels images of handwritten digits (0–9) (LeCun et al., 1998). We constructed a set dataset derived from the MNIST images and their corresponding digit labels. Each instance in our dataset was created by a set of randomly selected MNIST images of varying sizes, where the set label was obtained by applying an aggregation function to the set of associated image labels. The number of elements in each set, $n$, was sampled uniformly between $n_{min} = 2$ and $n_{max} = 16$. We employed ten different aggregation functions, including common measures of central tendency (e.g., mean, median), extremal values (max), summation, variance, and others. The task was to estimate the set labels. For each aggregating function we generated a new datasets, including training, validation and testing subsets; consisting of of $20 \times 10^3$, $2 \times 10^3$, and $3 \times 10^3$ instances respectively.

**Omniglot-sets dataset**  Omniglot is a collection of handwritten characters drawn from approximately 50 different alphabets, containing over 1,600 unique character classes, originally proposed for evaluation of a few-shot learning (Lake et al., 2015). Following a similar procedure to the MNIST-set dataset, we constructed variable-sized sets from the Omniglot dataset by randomly sampling images from the Omniglot collection. Each set was assigned a binary label vector indicating which of the 50 alphabets were represented in the set, based on the presence or absence of their characters. This framing naturally leads to a binary multi-label classification task, where the objective is to predict which alphabets appear in a given set. We sampled between $n_{min}$ and $n_{max}$ images, where the minimum set size was fixed at $n_{min} = 5$ was maximum was varying across $n_{max} \in \{10, 15, 20, 25\}$ to explore the effect of increasing the upper bound on the set size on the models' performance. For each value of $n_{max}$, we generated a new dataset comprising training, validation, and test subsets, consisting of of $20 \times 10^3$, $2 \times 10^3$, and $3 \times 10^3$ instances, respectively.

**ModelNet40 dataset**  ModelNet40 is a 3D object recognition dataset containing CAD models from 40 object categories, commonly used in point cloud classification tasks (Wu et al., 2015). Each object is represented as a point cloud consisting of a varying number of 3D coordinates. We uniformly subsampled each object to obtain a fixed number of points $n \in \{256, 512, 1024\}$. The resulting point clouds were then normalized to fit within a unit sphere centered at the origin, ensuring geometric consistency across all instances. The classification task involved predicting the correct object category from the point cloud representation. For each value of $n$, we generated a new dataset comprising training, validation, and test subsets, consisting of $20 \times 10^3$, $2 \times 10^3$, and $3 \times 10^3$ objects, respectively.

**QM9 dataset**  QM9 consists of computed geometric, energetic, electronic, and thermodynamic properties for 134,000 stable small organic molecules made up of C, H, O, N, and F (Ramakrishnan et al., 2014). These properties were obtained through computational simulation and, although they are close to experimentally measured values, the dataset cannot be strictly considered real-world. Each molecule consists of a variable number of atoms with associated 3D coordinates. The number of atoms ranges from 3 to 29, so the dataset to be viewed naturally as a set-valued input of varying cardinality. We conducted two independent prediction tasks: estimating the HOMO and LUMO energy levels, whose values we converted to milli-electron volts (meV) for improved numerical stability. For each task, we trained and evaluated the models using independent training, validation, and test splits, comprising $80\%$, $10\%$, and $10\%$ of the data, respectively.

## E.2 MODELS IMPLEMENTATION

All models, including our proposed QUANNs, its ablated variants, and the baseline methods, were implemented in PyTorch. The baselines were implemented based on the methodological descriptions provided in their respective papers and, when available, their official code repositories. Every effort was made to reproduce their original implementation faithfully and accurately.

To ensure a fair comparison across models, we standardized the encoder $\phi$ and estimator $\rho$ architectures used in all experiments (cf. Table 11). This consistency was maintained across all models, thereby isolating the impact of the aggregation mechanism as the primary variable under investigation.

Furthermore, to ensure comparable learning capacity between QUANN-1/-2 and the unary or binary aggregation-based baselines, we intentionally reduced the capacity of QUANNs' encoder and estimator networks. Specifically, one hidden layer was removed from each of these components in QUANNs. This adjustment led to all models having approximately the same number of learnable parameters, which mitigates any performance differences that could otherwise be attributed to model size (cf. Table 12).

## E.3 TRAINING & TESTING

In all experiments, each model was trained using only the training portion of the dataset. During training, we explored a range of learning rates in order to identify optimal training configurations. Following training, model selection was performed based on performance on the validation set. Specifically, for each model and learning rate configuration, we measured the validation loss and selected the trained instance that achieved the lowest validation error. Once the best model for each method was selected, final evaluation was conducted on the test portion of the dataset. The resulting test loss or, where applicable, another performance metric was recorded. Information about hyperparamters and loss functions selection are summarized in the Table 10.

## E.4 STATISTICAL EVALUATION

**Performance evaluation**  For each real-world dataset experiment, we performed 4 independent experimental replicates. For synthetic data experiments, we conducted 10 replicates. All reported values represent the mean and standard deviation computed across these experimental replicates. To assess whether one model significantly outperformed another, we adopted a commonly used informal heuristic: a model was considered to perform significantly better if the difference in mean performance exceeded the sum of the corresponding standard deviations.

**Win-loss matrices**  The models were additionally compared in pair-wise fashion and the results were conveyed as a win-loss matrix (Figure 4), summarizing the total number of outcomes where the model in the given row outperformed the model in the given column (comparing mean values only, irrespectively of the standard deviation), normalized by the total number of outcomes the models participated in (cf. Table 5).

**Binomial test**  To assess the statistical significance of the win proportions in the win-loss matrix, a one-tailed binomial test was conducted for each entry. The test was designed to evaluate whether the observed proportion of wins for a given model is greater than what would be expected by chance. The null hypothesis thus says that observed proportion of the wins $r$ is not greater than $0.5$ – the expected proportion under random chance:

$$H_0 : r \leq 0.5$$
$$H_1 : r > 0.5$$

The entries with the $p-value < 0.05$ were considered significant (*), and those with $p-value < 0.01$ were considered highly significant (**).

### E.5 COMPUTATIONAL RESOURCES

The experiments were performed across 4 NVIDIA RTX A6000 GPUs, and 28 AMD EPYC-Rome Processors, on Ubuntu 20.04.6 LTS (Focal Fossa). The project code is available at: `https://github.com/tomastokar/Quasi-Arithmetic-Neural-Networks`

## F SUPPLEMENTARY TABLES

Table 7: Performance of individual models in the additional MNIST-set aggregation experiments. All values represent the Mean Squared Error (MSE; lower is better), averaged across multiple experimental replicates, with standard deviations reported. Values highlighted by bold indicate best performance, while those highlighted by underline show the second best performance.

| | Unary | | | | | | | | | | Binary | | | | Non-Janossy | | | |
| | DeepSet | | PointNet | | NormDeepSet | | HPDS | | QUANN-1 | | SetTransformer | | QUANN-2 | | FSPool | | SlotAtt | |
| | Mean | Std | Mean | Std | Mean | Std | Mean | Std | Mean | Std | Mean | Std | Mean | Std | Mean | Std | Mean | Std |
|---|---|---|---|---|---|---|---|---|---|---|---|---|---|---|---|---|---|---|
| Geometric mean | 0.212 | 0.014 | 0.117 | 0.016 | 0.188 | 0.023 | 0.117 | 0.012 | 0.106 | 0.024 | 0.110 | 0.014 | **0.085** | 0.024 | 0.129 | 0.022 | 0.102 | 0.026 |
| LogMeanExp | 0.118 | 0.029 | 0.096 | 0.018 | 0.114 | 0.019 | 0.081 | 0.011 | **0.053** | 0.011 | 0.076 | 0.025 | 0.061 | 0.025 | 0.105 | 0.020 | 0.070 | 0.009 |
| harmonic | 0.079 | 0.016 | 0.082 | 0.004 | 0.070 | 0.007 | 0.063 | 0.005 | 0.030 | 0.004 | 0.037 | 0.010 | **0.024** | 0.005 | 0.060 | 0.016 | 0.033 | 0.008 |
| median | 0.665 | 0.042 | 0.849 | 0.053 | 0.680 | 0.041 | 0.376 | 0.028 | 0.218 | 0.051 | 0.382 | 0.046 | **0.137** | 0.021 | 0.417 | 0.085 | 0.259 | 0.021 |
| midrange | 0.059 | 0.015 | 0.026 | 0.002 | 0.062 | 0.004 | 0.030 | 0.004 | 0.026 | 0.008 | 0.057 | 0.009 | **0.024** | 0.007 | 0.044 | 0.017 | 0.035 | 0.003 |

Table 8: Summary of the point cloud aggregation functions used in the synthetic data experiments (cf. Section 7.1). Each function maps sets of vectors in $\mathbb{R}^d$ to a single representative vector ($\mathbb{R}^d \to \mathbb{R}^d$). Each function captures different statistical or geometric properties of the input point set.

| Type | Function Name | Expression |
|---|---|---|
| *Central tendencies* | Marginal median | $(\text{median}(x_{\cdot,j}))_{j=1}^d$ |
| | Geometric median | $\arg\min\limits_{z \in \mathbb{R}^d} \sum\limits_{i=1}^{n} \|\mathbf{x}_i - \mathbf{z}\|_1$ |
| | Medoid | $\arg\min\limits_{x_i \in \mathcal{X}} \sum\limits_{j=1}^{n} \|\mathbf{x}_i - \mathbf{x}_j\|_2$ |
| | Quadratic mean | $\left(\frac{1}{n} \sum\limits_{i=1}^{n} \mathbf{x}_i^2\right)^{1/2}$ |
| *Extremes* | Midpoint | $\frac{1}{2}(\mathbf{x}_i + \mathbf{x}_j), \quad \text{where } (i,j) = \arg\max\limits_{i,j} \|\mathbf{x}_i - \mathbf{x}_j\|_2$ |
| | VecMaxNorm | $\arg\max\limits_{x_i \in \mathcal{X}} \|\mathbf{x}_i\|_2$ |
| | Row max | $(\max(x_{\cdot,j}))_{j=1}^d$ |
| | LogSumExp | $\log\left(\sum\limits_{i=1}^{n} \exp(\mathbf{x}_i)\right)$ |
| *Moments* | Variance | $\frac{1}{n} \sum\limits_{i=1}^{n} (\mathbf{x}_i - \bar{\mathbf{x}})^2$ |
| | Skewness | $\frac{1}{n} \sum\limits_{i=1}^{n} \left(\frac{\mathbf{x}_i - \bar{\mathbf{x}}}{\sigma}\right)^3$ |

Table 9: Summary of the labels aggregation functions used in the aggregation-of-MNIST-images experiment. Each function operates on a set of scalars $\{x_1, x_2, \ldots, x_n\}$.

| Function Name | Formula |
|---|---|
| Mean | $\dfrac{1}{n} \sum_{i=1}^{n} x_i$ |
| Median | $\mathrm{median}(x_1, x_2, \ldots, x_n)$ |
| Mode | $\mathrm{mode}(x_1, x_2, \ldots, x_n)$ |
| Geometric Mean | $\left( \prod_{i=1}^{n} x_i \right)^{1/n}$ |
| Harmonic Mean | $\dfrac{n}{\sum_{i=1}^{n} \frac{1}{x_i}}$ |
| Log-Mean-Exp | $\log \left( \dfrac{1}{n} \sum_{i=1}^{n} e^{x_i} \right)$ |
| Midrange | $\dfrac{\min_i x_i + \max_i x_i}{2}$ |
| Variance | $\dfrac{1}{n} \sum_{i=1}^{n} (x_i - \bar{x})^2, \quad \text{where } \bar{x} = \frac{1}{n} \sum_{i=1}^{n} x_i$ |
| Maximum | $\max_i x_i$ |
| Sum | $\sum_{i=1}^{n} x_i$ |

Table 10: The hyper-parameters and loss function selection as used in our experiments.

| Dataset | Learning Rate | Batch Size | Epochs | Latent Dim | Loss | Performance Metric |
|---|---|---|---|---|---|---|
| Synthetic data | $1.0 \times 10^{-4}$ $5.0 \times 10^{-4}$ $1.0 \times 10^{-3}$ $5.0 \times 10^{-3}$ | 32 | 20 | 16 | MSE | MSE |
| MNIST-set | $1.0 \times 10^{-4}$ $5.0 \times 10^{-4}$ $1.0 \times 10^{-3}$ | 32 | 50 | 128 | MSE | MSE |
| Omniglot-set | $1.0 \times 10^{-4}$ $5.0 \times 10^{-4}$ $1.0 \times 10^{-3}$ | 32 | 50 | 128 | BCE | Ballanced ACC |
| ModelNet40 | $1.0 \times 10^{-4}$ $5.0 \times 10^{-4}$ $1.0 \times 10^{-3}$ | 32 | 50 | 128 256 | MSE | ACC |

Table 11: Summary of encoder $\psi$ and predictor $\rho$ architectures used for individual modalities across the given datasets. Note, $dim$ indicates latent space dimension.

| Dataset | Encoder $\phi$ | Estimator $\rho$ |
|---|---|---|
| Synthetic | Input: $\mathbb{R}^{16}$
FC 128 + ReLU
FC $dim$ | Input: $\mathbb{R}^{dim}$
FC 128 + ReLU
FC 16 |
| MNIST-set | Input: $\mathbb{R}^{1\times28\times28}$
Conv 16, kernel $3\times3$, stride 1, pad 1 + ReLU
Conv 32, kernel $3\times3$, stride 1, pad 1 + ReLU
Conv 64, kernel $3\times3$, stride 1, pad 1 + ReLU
MaxPool2d
FC $dim$ | Input: $\mathbb{R}^{dim}$
FC 256 + ReLU
FC 256 + ReLU
FC 1 |
| Omniglot-set | Input: $\mathbb{R}^{1\times28\times28}$
Conv 16, kernel $3\times3$, stride 1, pad 1 + ReLU
Conv 32, kernel $3\times3$, stride 1, pad 1 + ReLU
Conv 64, kernel $3\times3$, stride 1, pad 1 + ReLU
MaxPool2d
FC $dim$ | Input: $\mathbb{R}^{dim}$
FC 256 + ReLU
FC 256 + ReLU
FC 50 |
| ModelNet40 | Input: $\mathbb{R}^{3}$
FC 256 + ReLU
FC 256 + ReLU
FC $dim$ | Input: $\mathbb{R}^{dim}$
FC 256 + ReLU
FC 256 + ReLU
FC 40 |

Table 12: Number of trainable parameters per model across all experiments.

| | Synthetic | MNIST | MNIST-pretrained | Omniglot | ModelNet40 | QM9 |
|---|---|---|---|---|---|---|
| Ablation 1 | 8480 | | | | | |
| Ablation 2 | 15776 | | | | | |
| Ablation 3 | 11696 | | | | | |
| DeepSet | | 196225 | 99073 | 208818 | 208808 | 200065 |
| NormDeepSet | | 196225 | 99073 | 208818 | 208808 | 200065 |
| HPDS | | 196226 | 99074 | 208819 | 208809 | 200066 |
| **QUANN-1** | 11696 | 196737 | 99585 | 209330 | 209320 | 134785 |
| SetTransformer | | 263553 | 166401 | 269874 | 271144 | 267393 |
| **QUANN-2** | | 246657 | 149505 | 252978 | 320552 | 316801 |
| FSPool | | 196245 | 99093 | 208838 | 208036 | 134293 |
| SlotAtt | | 281217 | 184065 | 287538 | 288808 | 285057 |
| LAF | | 197761 | 100609 | 210354 | 210344 | 201601 |

# G SUPPLEMENTARY FIGURES

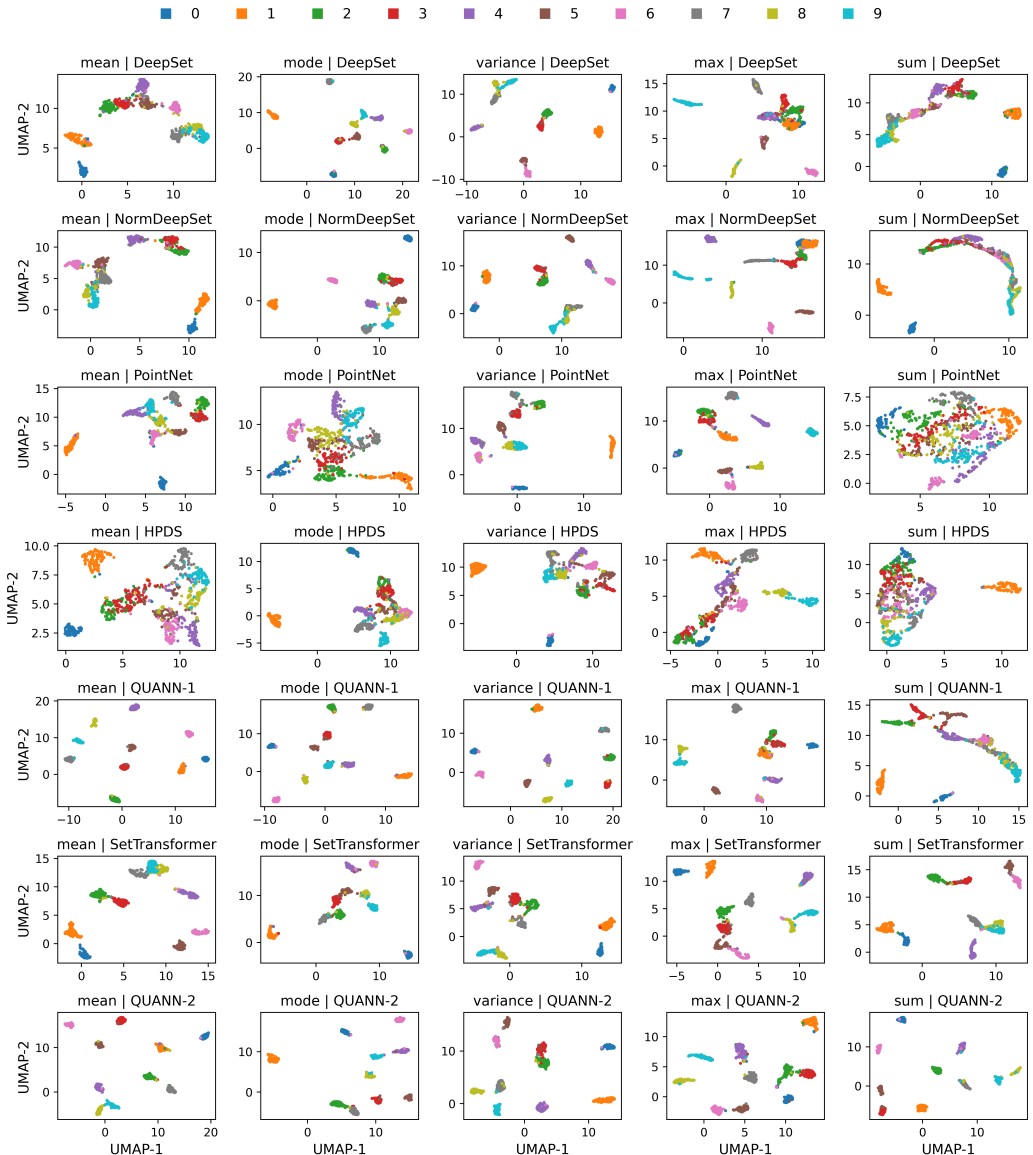

Figure 7: UMAP projections of the MNIST test set embeddings obtained from the encoder networks ($\phi$) trained under different Janossy methods to approximate various aggregation functions. Each point corresponds to an image, colored by its digit class label. Notably, only the encoders trained via SetTransformer, QUANN-1 (unary pooling), and QUANN-2 (binary pooling) produce embeddings that exhibit clear class-wise separation over all tasks, highlighting their superior ability to organize inputs in a semantically meaningful latent space.

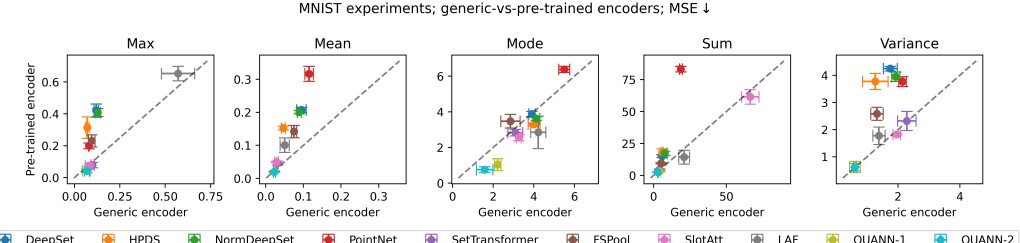

Figure 8: Set function models trained with a generic vs. a fixed, pre-trained MNIST image encoder. Subplots correspond to a different aggregating functions, and show the model performance (MSE, lower is better), with error bars indicating standard deviations across experimental replicates. Only QUANN-1 and -2 retain, or improve, their performance despite fixed encoder (points close to, or below, the identity line), whereas baselines experience a significant performance drop (points above the identity lines); despite comparable number of trainable parameters (cf. Table 12).

## H    PROOFS

**Corollary H.1** (Permutation invariance of Quasi-arithmetic Neural Networks). *Let $\mathbf{X}$ be a finite set of elements as described in Section 2 , and let $\hat{F}(\mathbf{X})$ be a set function approximation as described in Eq. 4. Then, $\hat{F}$ is permutation invariant with respect to the elements of the input set $\mathbf{X}$.*

*Proof.* For any permutation $\pi$ of the indices $\{1, \ldots, n\}$, we have:

$$\hat{F}(X') = \rho\left(\phi^{-1}\left(\frac{1}{n}\sum_{i=1}^{n}\psi(\phi(x_{\pi(i)}))\right)\right) = \rho\left(\phi^{-1}\left(\frac{1}{n}\sum_{i=1}^{n}\psi(\phi(x_i))\right)\right) = \hat{F}(X),$$

since summation is invariant under permutation. Thus, $\hat{F}$ is permutation invariant. $\square$

**Corollary H.2** (Kolmogorov Mean with Linear Generator is Equal to Arithmetic Mean). *Let $M_\psi(X)$ be Kolmogorov mean as defined in Equation 3. If $\psi : \mathbb{R} \to \mathbb{R}$ is a linear function of the form $\psi(x) = w_1 x + w_2$ for constants $w_1 \neq 0$ and $w_2 \in \mathbb{R}$, then $M_\psi$ is equal to the arithmetic mean.*

*Proof.* Let $\psi(x) = w_1 x + w_2$, with $w_1 \neq 0$, so that $\psi$ is invertible with inverse $\psi^{-1}(y) = (y - w_2)/w_1$. The Kolmogorov mean is then computed as:

$$M_\psi(x_1, \ldots, x_n) = \psi^{-1}\left(\frac{1}{n}\sum_{i=1}^{n}\psi(x_i)\right) = \psi^{-1}\left(\frac{1}{n}\sum_{i=1}^{n}(w_1 x_i + w_2)\right).$$

Simplifying the expression inside the inverse:

$$\frac{1}{n}\sum_{i=1}^{n}(w_1 x_i + w_2) = w_1\left(\frac{1}{n}\sum_{i=1}^{n}x_i\right) + w_2.$$

and apply $\psi^{-1}$:

$$\psi^{-1}\left(w_1\left(\frac{1}{n}\sum_{i=1}^{n}x_i\right) + w_2\right) = \frac{w_1\left(\frac{1}{n}\sum_{i=1}^{n}x_i\right) + w_2 - w_2}{w_1} = \frac{1}{n}\sum_{i=1}^{n}x_i.$$

Therefore,

$$M_\psi(X) = \frac{1}{n}\sum_{i=1}^{n}x_i.$$

$\square$

*Proof of therorem 5.1.* Let's decompose $\mathcal{U}$ as the following union:

$$\mathcal{U} = \mathcal{U}_{\psi=\mathbb{I}} \cup \mathcal{U}_{\psi\neq\mathbb{I}}.$$

where $\mathcal{U}_{\psi\neq\mathbb{I}}$ denotes set of functions that can be uniformly approximated by QUANNs of the form equation 4 with $\psi = \mathbb{I}$, while $\rho, \phi$ are arbitrary neural networks.

By the representation result of  Bueno & Hylton (2021), every permutation-invariant set function $F : \mathcal{P}_f(X) \to \mathcal{Y}$ that is uniformly continuous with respect to the Wasserstein metric can be uniform approximated by a function of the form

$$F(X) = \rho\left(\frac{1}{|X|}\sum_{x\in X}\phi(x)\right),$$

where $\rho$ and $\phi$ are arbitrary chosen neural networks.

This representation corresponds to the QUANN architecture specialized to $\psi = \mathbb{I}$ (i.e. the quasi-arithmetic mean reduces to a sum or mean when $\psi$ is the identity) and so $\mathcal{U}_{\psi=\mathbb{I}} = \mathcal{U}_W$

From the above we obtain that $\mathcal{U} = \mathcal{U}_{\psi=\mathbb{I}} \cup \mathcal{U}_{\psi\neq\mathbb{I}} = \mathcal{U}_W \cup \mathcal{U}_{\psi\neq\mathbb{I}}$ and therefore:

$$\mathcal{U} \supseteq \mathcal{U}_W,$$

$\square$

**Proposition H.3** (Approximation of Mean-Decomposable Set Function). *Let $X = \{x_1, \ldots, x_n\}$ be a finite set as defined in Section 2, and consider a mean-decomposable set function F(x) of the form:*

$$F(X) = a\left(\frac{1}{n}\sum_{i=1}^{n} b(x_i)\right),$$

*where:*

- $b : \mathcal{X} \to \mathbb{R}^d$ *is a continuous function;*

- $a : \mathbb{R}^d \to \mathbb{R}^{d'}$ *is a Lipschitz continuous function with Lipschitz constant $L_a$, for some $d, d' \in \mathbb{N}$.*

*Let $\hat{F}(X)$ denote the approximation of $F(X)$ produced by the model described in Section 5, which constructs:*

- *a continuous function $\phi \approx b$,*

- *a generator $\psi$ approximating any linear map $x \mapsto w_1 x + w_2$, and*

- *a continuous function $\rho \approx a$.*

*Then, for any $\epsilon > 0$, if $\psi$ approximates a linear function sufficiently well and both $\phi$ and $\rho$ approximate $b$ and $a$, respectively, within corresponding tolerances, the approximation error is bounded as:*

$$\|F(X) - \hat{F}(X)\| \leq \epsilon.$$

*Proof.* Let $\mu_b := \frac{1}{n}\sum_{i=1}^{n} b(x_i)$ and $\mu_{\psi,\phi} := \psi^{-1}\left(\frac{1}{n}\sum_{i=1}^{n}\psi(\phi(x_i))\right)$. The approximation error can be expressed as:

$$\|\hat{F}(X) - F(X)\| = \|a(\mu_b) - \rho(\mu_{\psi,\phi})\|$$

By the triangle inequality and the Lipschitz continuity of $a$, along with the universal approximation property of $\rho$ (i.e., $\|a(z) - \rho(z)\| \leq \epsilon_\rho$ for any $z$), we have:

$$\|\hat{F}(X) - F(X)\| = \|a(\mu_b) - \rho(\mu_{\psi,\phi})\| \tag{15}$$

$$\leq \|a(\mu_b) - a(\mu_{\psi,\phi})\| + \|a(\mu_{\psi,\phi}) - \rho(\mu_{\psi,\phi})\| \tag{16}$$

$$\leq L_a \|\mu_b - \mu_{\psi,\phi}\| + \epsilon_\rho, \tag{17}$$

where $L_a$ is the Lipschitz constant of $a$.

We now focus on bounding $\|\mu_b - \mu_{\psi,\phi}\|$. By definition:

$$\|\mu_b - \mu_{\psi,\phi}\| = \left\|\frac{1}{n}\sum_{i=1}^{n} b(x_i) - \psi^{-1}\left(\frac{1}{n}\sum_{i=1}^{n}\psi(\phi(x_i))\right)\right\|.$$

From Corollary H.2, if $\psi$ approximates a linear function $\psi(x) \approx w_1 x + w_2$ such that $\|\psi(x) - (w_1 x + w_2)\| \leq \epsilon_\psi$, then in the limit as $\epsilon_\psi \to 0$, the Kolmogorov mean reduces to the arithmetic mean:

$$\lim_{\epsilon_\psi \to 0} \mu_{\psi,\phi} = \frac{1}{n}\sum_{i=1}^{n}\phi(x_i).$$

Using this and the universal approximation property of $\phi$, i.e., $\|b(x) - \phi(x)\| \leq \epsilon_\phi$, we can bound:

$$\lim_{\epsilon_\psi \to 0} \|\mu_b - \mu_{\psi,\phi}\| = \left\|\frac{1}{n}\sum_{i=1}^{n} b(x_i) - \frac{1}{n}\sum_{i=1}^{n}\phi(x_i)\right\| \tag{18}$$

$$= \left\|\frac{1}{n}\sum_{i=1}^{n}(b(x_i) - \phi(x_i))\right\| \tag{19}$$

$$\leq \frac{1}{n}\sum_{i=1}^{n}\|b(x_i) - \phi(x_i)\| \tag{20}$$

$$\leq \epsilon_\phi. \tag{21}$$

Substituting into the earlier inequality:

$$\lim_{\epsilon_\psi \to 0} \|\hat{F}(X) - F(X)\| \le L_a \epsilon_\phi + \epsilon_\rho.$$

Finally, since both $\epsilon_\phi$ and $\epsilon_\rho$ can be made arbitrarily small, we can write:

$$\lim_{\epsilon_\psi \to 0} \|\hat{F}(X) - F(X)\| \le \epsilon,$$

for an arbitrarily small $\epsilon > 0$.

$\square$

**Proposition H.4** (Approximation of Max-Decomposable Set Function). *Let $X = \{x_1, \ldots, x_n\}$ be a finite set as defined in Section 2, and consider a mean-decomposable set function F(x) of the form:*

$$F(X) = a \left( \max_{x \in X} b(x) \right),$$

*where:*

- $b : \mathcal{X} \to \mathbb{R}^d$ *is a continuous function;*

- $a : \mathbb{R}^d \to \mathbb{R}^{d'}$ *is a Lipschitz continuous function with Lipschitz constant $L_a$, for some $d, d' \in \mathbb{N}$.*

*Let $\hat{F}(X)$ denote the approximation of $F(X)$ produced by the model described in Section 5, which constructs:*

- *a continuous function $\phi \approx b$,*

- *a generator $\psi$ approximating any exponential function $x \mapsto \exp(wx)$, s.t. $w > 0$, and*

- *a continuous function $\rho \approx a$.*

*Then, for any $\epsilon > 0$, if $\psi$ approximates an exponential family function sufficiently well and both $\phi$ and $\rho$ approximate $b$ and $a$, respectively, within corresponding tolerances, the approximation error is bounded as:*

$$\|F(X) - \hat{F}(X)\| \le L_a \log(n) + \epsilon.$$

*Proof.* Let $M_b := \max_{x \in X} b(x)$ and $\mu_{\psi, \phi} := \psi^{-1}(\frac{1}{n} \sum_{i=1}^n \psi(\phi(x_i)))$. The approximation error can be expressed as:

$$\|F(X) - \hat{F}(X)\| = \left\| a(M_b) - \rho(\mu_{\psi, \phi}) \right\| \tag{22}$$

By the triangle inequality and the Lipschitz continuity of $a$, along with the universal approximation property of $\rho$ (i.e., $\|a(z) - \rho(z)\| \le \epsilon_\rho$ for any $z$), we have:

$$\|F(X) - \hat{F}(X)\| = \|a(M_b) - \rho(\mu_{\psi, \phi})\| \tag{23}$$
$$\le \|a(M_b) - a(\mu_{\psi, \phi})\| + \|a(\mu_{\psi, \phi}) + \rho(\mu_{\psi, \phi})\| \tag{24}$$
$$\le L_a \|M_b - \mu_{\psi, \phi}\| + \epsilon_\rho \tag{25}$$

where $L_a$ is the Lipschitz constant of $a$.

We now focus on the bounding $\|M_b - \mu_{\psi, \phi}\|$. By definition:

$$\|M_b - \mu_{\psi, \phi}\| = \left\| M_b - \psi^{-1} \left( \frac{1}{n} \sum_{x \in X} \psi(\phi(x)) \right) \right\| \tag{26}$$

When $\psi$ approximates an exponential function $\psi(x) \approx \exp(wx)$ such that $\|\psi(x) - \exp(wx)\| \leq \epsilon_\psi$ then in the limit as $\epsilon_\psi \to 0$, the above norm can be computed as:

$$\lim_{\epsilon_\psi \to 0} \|M_b - \mu_{\psi,\phi}\| = \left\| M_b - \frac{1}{w} \log \left( \frac{1}{n} \sum_{i=i}^{n} \exp(w\phi(x)) \right) \right\| \tag{27}$$

$$= \left\| \frac{1}{w} \log \left( \frac{1}{n} \sum_{i=i}^{n} \exp(w\phi(x)) \right) - M_b \right\| \tag{28}$$

$$= \left\| \frac{1}{w} \log \left( \frac{1}{n} \sum_{i=i}^{n} \exp(w\phi(x)) \right) - \frac{1}{w} \log \left( \exp(wM_b) \right) \right\| \tag{29}$$

$$= \left\| \frac{1}{w} \log \left( \frac{1}{n} \sum_{i=i}^{n} \exp(w\phi(x) - wM_b) \right) \right\| \tag{30}$$

$$\tag{31}$$

We will now establish the lower bound on the sum inside the logarithm. We will leverage the approximation boundary $\|\phi(x) - b(x)\| \leq \epsilon_\phi$; and that $b(x) = M_b$ for at least one $x \in X$.

$$\frac{1}{n} \sum_{i=1}^{n} \exp(w\phi(x) - wM_b) = \frac{1}{n} \sum_{i=1}^{n} \exp(w(\phi(x) - M_b)) \tag{32}$$

$$\geq \frac{1}{n} \sum_{i=1}^{n} \exp(w(b(x) - M_b - \epsilon_\phi)) \tag{33}$$

$$\geq \frac{1}{n} \sum_{x \in X,\, b(x) = M_b} \exp(w(b(x) - M_b - \epsilon_\phi)) \tag{34}$$

$$\geq \frac{1}{n} \exp(-w\epsilon_\phi) \tag{35}$$

Given this bound we establish the upper bound on the limit of the above norm:

$$\frac{1}{n} \sum_{i=1}^{n} \exp(w\phi(x) - wM_b) \geq \frac{1}{n} \exp(-w\epsilon_\phi) \implies \lim_{\epsilon_\psi \to 0} \|M_b - \mu_{\psi,\phi}\| \leq \frac{1}{w} \log(n) + \epsilon_\phi \tag{36}$$

Substituting into the earlier inequality:

$$\lim_{\epsilon_\psi \to 0} \|\hat{F}(X) - F(X)\| \leq L_a \left( \frac{1}{w} \log(n) + \epsilon_\phi \right) + \epsilon_\rho \tag{37}$$

Finally, since both $\epsilon_\phi$ and $\epsilon_\rho$ can be made arbitrarily small, we can write:

$$\lim_{\epsilon_\psi \to 0} \|\hat{F}(X) - F(X)\| \leq L_a \frac{1}{w} \log(n) + \epsilon \tag{38}$$

for an arbitrarily small $\epsilon > 0$.

$\square$

**Proposition H.5** (Approximation of Sum-Decomposable Set Function). *Let $X = \{x_1, \ldots, x_n\}$ be a finite set as defined in Section 2, and consider a sum-decomposable set function F(x) of the form:*

$$F(X) = a \left( \sum_{i=1}^{n} b(x_i) \right),$$

*where:*

- *$b : \mathcal{X} \to \mathbb{R}^d$ is continuous function bounded by the interval $(B_0, B_1)$;*

- $a : \mathbb{R}^d \rightarrow \mathbb{R}^{d'}$ *is a Lipschitz continuous function with Lipschitz constant* $L_a$, *for some* $d, d' \in \mathbb{N}$.

*Let* $\hat{F}(X)$ *denote the approximation of* $F(X)$ *produced by the model described in Section 5, which constructs:*

- *a continuous function* $\phi \approx wb$, $w > 0$,

- *a continuous function* $\rho \approx a$.

*Then, for any* $\epsilon > 0$, *if* $\psi$ *approximates any continuous invertible function and both* $\phi$ *and* $\rho$ *approximate functions* $wb$ *and* $a$, *respectively, within corresponding tolerances, the approximation error is bounded as:*

$$\|F(X) - \hat{F}(X)\| \le L_a(nB_1 - wB_0) + \epsilon$$

*Proof.* Let $S_b := \sum_{i=1}^n b(x_i)$ and $\mu_{\psi,\psi} = \psi^{-1}(\frac{1}{n}\sum_{i=1}^n \psi(\phi(x)))$. The approximation error can be expressed as:

$$\|F(X) - \hat{F}(X)\| = \left\|a(S_b) - \rho(\mu_{\psi,\phi})\right\| \tag{39}$$

Same as in the proofs of Propositions H.3 and H.4:

$$\|F(X) - \hat{F}(X)\| = \|a(S_b) - \rho(\mu_{\psi,\phi})\| \tag{40}$$
$$\le \|a(Sb) - a(\mu_{\psi,\phi})\| + \|a(\mu_{\psi,\phi}) + \rho(\mu_{\psi,\phi})\| \tag{41}$$
$$\le L_a\|S_b - \mu_{\psi,\phi}\| + \epsilon_\rho \tag{42}$$

Since $wb(x) - \epsilon_\phi < \phi(x) < wb(x) + \epsilon_\phi$ and $B_0 < b(x) < B_1$ then leveraging that Kolmogoov mean is neither larger than the largest input, nor smaller than the smallest input (cf. Section 4), we obtain: $wB_0 - \epsilon_\phi < \mu_{\psi,\phi} < wB_1 + \epsilon_\phi$.

Given this bound we establish the upper bound on the above norm:

$$\|S_b - \mu_{\psi,\phi}\| = \left\|S_b - \psi^{-1}\left(\frac{1}{n}\sum_{x \in X}\psi(\phi(x))\right)\right\| \tag{43}$$
$$\le \|nB_1 - wB_0 + \epsilon_\phi\| \tag{44}$$

Substituting into the earlier inequality:

$$\|\hat{F}(X) - F(X)\| \le L_a((nB_1 - wB_0) + \epsilon_\phi) + \epsilon_\rho \tag{45}$$

Finally, since both $\epsilon_\phi$ and $\epsilon_\rho$ are arbitrarily small, we can write:

$$\|\hat{F}(X) - F(X)\| \le L_a(nB_1 - wB_0) + \epsilon \tag{46}$$

$\square$

**Proposition H.6** (Expected value of the Approximation of Sum-Decomposable Set Function). *(Extending the Proposition H.5) Assuming cardinality* $n$ *of the set* $X$ *follows independent distribution* $p(n)$, *with* $\bar{n}$ *being its expected value, then, for any* $\epsilon > 0$, *if* $\psi$ *approximates any invertible continuous function and both* $\phi$ *and* $\rho$ *approximate functions* $\bar{n}B_1/B_0 \cdot b$ *and* $a$, *respectively, within corresponding tolerances, the expected value of the approximation error is bounded as:*

$$\mathbb{E}\big[\|\hat{F}(X) - F(X)\|\big] \le \epsilon \tag{47}$$

*Proof.* From the previous proposition we obtain that:

$$\|\hat{F}(X) - F(X)\| \le L_a(nB_1 - wB_0) + \epsilon \tag{48}$$

The expected value of the approximation error is:

$$\mathbb{E}\big[\|\hat{F}(X) - F(X)\|\big] \le L_a(\bar{n}B_1 - wB_0) + \epsilon \tag{49}$$

When the scale $w$, learned by $phi$, approximates the $\bar{n}B_1/B_0$: $w \approx \bar{n}B_1/B_0 \cdot b$, such that $\|w - \bar{n}B_1/B_0 \cdot b\| < \epsilon_w$ then in the limit $\epsilon_w \to 0$, the above expected value is bounded by:

$$\lim_{\epsilon_w \to 0} \mathbb{E}\big[\|\hat{F}(X) - F(X)\|\big] \le L_a \epsilon_w + \epsilon \tag{50}$$

Finally, since both $\epsilon$ and $\epsilon_w$ are arbitrarily small, we can write:

$$\lim_{\epsilon_w \to 0} \mathbb{E}\big[\|\hat{F}(X) - F(X)\|\big] \le \epsilon \tag{51}$$

for an arbitrarily small $\epsilon > 0$.

$\square$

