# OpenReview forum: "Improving Set Function Approximation with Quasi-Arithmetic Neural Networks"
_ICLR.cc/2026/Conference — ICLR 2026 Poster_

### Official Review · Reviewer_3i4p · 2025-10-15

**Soundness:** 2
**Presentation:** 3
**Contribution:** 2
**Rating:** 4
**Confidence:** 5

**Summary:**

The paper proposed a novel trainable aggregation function for sets, called Neuralized Kolmogorov Mean, and a quasi-arithmetic neural networks which includes the Neuralized Kolmogorov Mean in it.  The authors explore both unary and binary (similar to set attention) QANNs. The main contributions of the paper are:
1. A novel aggregation function for sets - the Neuralized Kolmogorov Mean.
2. Theoretical justification of its soundness, including proofs of permutation invariance and universal approximation properties.
3. The results show some consistent although sometimes slight benefit over other methods, with the results about transfer learning to and from classification tasks helping to support the author’s claims of better structured latent spaces.

**Strengths:**

The main strengths of the paper include:
- the idea is quite simple and easily explained, and the paper is clear with its intentions, methods and outcomes.
- a novel learnable aggregations function
- theoretical justifications

**Weaknesses:**

The mean weakness of the paper are:
- poor comparison against other learnable aggregation functions
- the majority of the experiments relies on semi-synthetic datasets and very simple datasets such as Omniglot and MNIST, so there is no evidence that the benefits observed would translate into a more meaningful task.
- the accuracy on ModelNet40 for DeepSets is quite low compared to the original paper (82% for 100 points, and 90% for 5,000 points), but the authors report ~65% (see Table 5).

All the weakness are expanded and further explained in the question section below.

**Questions:**

The methodological and novelty part of the paper is convincing, but unfortunately the experimental sections is somewhat weak. I would like to mention that there are many methods in the literature, and it would be unrealistic to expect any paper to include all possible competitors. However, it is needed to compare against the most relevant and closely related methods, such as [1] and [2].

**Methodology**.
1. As it stands, the paper seems to claim (see for e.g. the abstract) that the learnable function is entirely novel. This is inaccurate, as other methods such as PNA[1] and LAF[2] (among others) have already introduced learnable aggregation functions to address the problem of having fixed aggregation functions. Why did the authors not position their proposed aggregation function in relation to [1] and [2]?

**Experiments**.
1. Comparisons with closely related works (e.g. [1], and [2]) are missing. Is there a justification as of why those methods were not included in the experimental evaluation?
2. The reported accuracies for DeepSets[3] are much lower compared to the original work. The paper reports ~65% (see Table 5), while the original paper achieved 82% for 100 points and 90% for 5,000 points. Why is there such a discrepancy? Also comparisons agains other methods are missing as previously mentioned. What is the rationale for their exclusion?
3. The authors mention a potential application for aggregations is in graph learning tasks. There is quite some literature on neural aggregation for graph level tasks (in the context of combing node representations after message passing) and use of e.g. set transformers in that context can give performance uplift. I see no reason why this method would not work in that context, and I would encourage the authors to investigate this aspect, at least briefly. The large number of practical tasks that have graph representations would help increase the potential impact of this work.

**Results**.
1. The MNIST-Sets experiments are quite related to those presented in [2]. Why was this comparison not included?
2. The results for the MNIST-Sets experiments (Table 5, "sum" task) are difficult to interpret relative to DeepSets, which reported an accuracy (not MSE) of over 60% for sets containing 50 images. It is unclear how MSE is as a meaningful metric for this task, since the aggregation output (a sum) is an integer. What would the accuracy be for QUANN-1 and QUANN-2 on the sum task?
3. It is not clear why ablation 2 performs so much worse than Ablation 1, when it is the same as Ablation1 but includes more parameters? Is there any explanation?

**Minor issues**.
1. Table 5 is too small and it should be broken perhaps in two parts. Experiments from the different subsections are summarized in the same Table which makes the reading a bit confusing.
2. Research questions section: does the proposed approach [learn] a.
3. Missing closing parenthesis at the end of the Sum decomposition section

[1] Corso, Gabriele, et al. "Principal neighbourhood aggregation for graph nets." Advances in neural information processing systems 33 (2020): 13260-13271.

[2] Pellegrini, Giovanni, et al. "Learning Aggregation Functions." IJCAI. International Joint Conferences on Artificial Intelligence Organization, 2021.

[3] Zaheer, Manzil, et al. "Deep sets." Advances in neural information processing systems 30 (2017).

---

> ### Author Response · Authors · 2025-11-21
> **Response to Reviewer 3i4p**
>
> **1. Poor comparison against other learnable aggregation functions**
>
> We extended our experiments by adding an additional baseline – LAF (Pellegrini et al. 2021) – as suggested by the reviewers. There are currently 4 methods that can be loosely characterized as relying on a learnable aggregation function: HDFS (unary Janossy pooling), LAF, SlotAtt (slot attention attention) and also FSPool (optimal permutation learning). Our results show that QUANNs outperform all of these methods (cf. revised Table 5).
>
> **2. The majority of the experiments relies on semi-synthetic datasets and very simple datasets such as Omniglot and MNIST, so there is no evidence that the benefits observed would translate into a more meaningful task.**
>
> The datasets used in our study were selected to align with those commonly adopted in prior work on permutation-invariant learning, e.g., MNIST-Sets was used in Zaheer et al. (2017, DeepSets) and Pellegrini et al. (2021, LAF), Omniglot in Lee et al. (2019, SetTransformer), and ModelNet40 in several studies. In addition, the MNIST and Omniglot experiments were important for evaluating the transfer-learning capabilities of our models relative to the baselines.
>
> To further strengthen our empirical evaluation, we also incorporated the real-world QM9 dataset (Ramakrishnan et al., 2014).
>
> **3. The accuracy on ModelNet40 for DeepSets is quite low compared to the original paper (82% for 100 points, and 90% for 5,000 points), but the authors report ~65% (see Table 5).**
>
> There are important differences between our ModelNet40 experiments and those in Zaheer et al. The key distinctions are that Zaheer et al. applied zero-mean, unit-variance rescaling and fixed set sizes (100, 1000, 5000), whereas we used unit-sphere rescaling and variable-size sets (see Supplementary Section F.1). The setup in Zaheer et al. is particularly favorable to sum-based DeepSets: zero-centered inputs and fixed cardinalities mitigate the expansive behavior of the sum aggregator and allow it to behave effectively like a mean. We believe these differences account for the lower accuracy we report.
> Moreover, our results are consistent with those observed in Qi et al. 2017b (PointNet++), where unit-sphere rescaling was also used.
>
> **4. As it stands, the paper seems to claim (see for e.g. the abstract) that the learnable function is entirely novel. This is inaccurate, as other methods such as PNA[1] and LAF[2] (among others) have already introduced learnable aggregation functions to address the problem of having fixed aggregation functions. Why did the authors not position their proposed aggregation function in relation to [1] and [2]?**
>
> We apologize for the overly bold statements. We revised the abstract and introduction to tone down our language (lines 15-19 and 54-55). Furthermore, please see our response no 5, below.
>
> **5. Comparisons with closely related works (e.g. [1], and [2]) are missing. Is there a justification as to why those methods were not included in the experimental evaluation?**
>
> Both suggested methods implement learnable aggregation by combining multiple input-set aggregates (e.g., power means, max, min) via fixed-form functions. As such, they can be characterized as non-Janossy methods, which are already represented in our experiments by more complex alternatives such as FSPool and Slot Attention. Nonetheless, we have expanded our evaluation by including LAF aggregation (Pellegrini et al., 2021) as an additional baseline. The results show that LAF performs worse than QUANNs across nearly all tasks (cf. revised Table 5).
>
> **6. The reported accuracies for DeepSets[3] are much lower compared to the original work. The paper reports ~65% (see Table 5), while the original paper achieved 82% for 100 points and 90% for 5,000 points. Why is there such a discrepancy? Also comparisons against other methods are missing as previously mentioned. What is the rationale for their exclusion?**
>
> Please see our response no 3 and 5.

---

> > ### Author Response · Authors · 2025-11-21
> > **Response to Reviewer 3i4p - cont.**
> >
> > **7. The MNIST-Sets experiments are quite related to those presented in [2]. Why was this comparison not included?**
> >
> > We have expanded our experiments to include this comparison. Overall, the results are consistent with those reported by Pellegrini et al. 2021 (Figure D.1). The only notable exception is the approximation of the sum, where differences arise due to the different ranges of set cardinalities used in our experiments versus theirs.
> >
> > **8. The results for the MNIST-Sets experiments (Table 5, "sum" task) are difficult to interpret relative to DeepSets, which reported an accuracy (not MSE) of over 60% for sets containing 50 images. It is unclear how MSE is as a meaningful metric for this task, since the aggregation output (a sum) is an integer. What would the accuracy be for QUANN-1 and QUANN-2 on the sum task?**
> >
> > We used MSE in evaluation of the sum-function approximation to remain consistent with the evaluation of the remaining aggregation functions. Moreover, while the sum produces an integer, it is still a quantity, not a categorical label, making MSE or a similar regression loss more appropriate. Since we used MSE also as a training objective, the assessing accuracy of QUANNs wouldn’t be very meaningful, as they were trained for regression.
> >
> > **9. It is not clear why ablation 2 performs so much worse than Ablation 1, when it is the same as Ablation1 but includes more parameters? Is there any explanation?**
> >
> > In Section 5.3, we argue that allocating part of the model’s capacity to the invertible function \psi effectively acts as an inner regularization, improving the learning process. Without this invertible component, Ablation 2 becomes over-capacitated and prone to overfitting, leading to worse generalization compared to the smaller Ablation 1.
> >
> > **10. Table 5 is too small and it should be broken perhaps in two parts. Experiments from the different subsections are summarized in the same Table which makes the reading a bit confusing.**
> >
> > We have split Table 5 into two separate tables: one for unary Janossy methods and another for binary and non-Janossy methods. Unfortunately, due to space limitations, we could not further separate the tables by individual experiment without moving results to the Appendix. We believe that the current layout makes the results clearer and easier to interpret.
> >
> > **11. Research questions section: does the proposed approach [learn] a.**
> >
> > Corrected as per reviewer suggestion. Thank you.
> >
> > **12. Missing closing parenthesis at the end of the Sum decomposition section**
> >
> > The missing parenthesis at the end of the paragraph was corrected.

---

> ### Comment · Reviewer_3i4p · 2025-11-25
> **Response to rebuttal**
>
> Thank for answering all of my questions and for conducting an additional experiment on QM9. However, I have still some concerns.
>
> 1. Zaheer et al. use max pooling for ModelNet40, not the sum (see Appendix H). Their preprocessing is also very simple, so it is difficult for me to reconcile a 25-point less accuracy, compared to your method.
>
> 2. The LAF aggregation experiments in Table 4 and 5 are not convincing. In principle, LAF can learn sum, mean, max, and other permutation invariant operations. For example in Table 4, it is surprising to me that DeepSets achieve an accuracy of 0.524 while LAF performs much worse at 0.156.
>
> 3. (minor) The additional experiments on QM9 are a good addition but I wouldn't call it a real world dataset as its properties are obtained from simulations (see the Background and Summary sections of [1] or https://www.tensorflow.org/datasets/catalog/qm9), although they are close to experimental accuracy. Moreover, QM9 is a subset of GDB-11, which was artificially generated to enumerate all possible molecules containing up to 11 atoms of H, C, N, O, and F.
>
> [1] Ramakrishnan, Raghunathan, et al. "Quantum chemistry structures and properties of 134 kilo molecules." Scientific data 1.1 (2014): 1-7.

---

> > ### Author Response · Authors · 2025-11-27
> > **Response to additional comments by reviewer 3i4p**
> >
> > **1. Zaheer et al. use max pooling for ModelNet40, not the sum (see Appendix H). Their preprocessing is also very simple, so it is difficult for me to reconcile a 25-point less accuracy, compared to your method.**
> >
> > To assess the correspondence between our results and those performed by Zaheer at al. we performed the following additional experiment:
> >
> > We implemented the permutation-equivariant variant of DeepSets as described in Zaheer et al. (Section 3.1, Equivariant Model, and Appendix C). We then assessed this model under different pooling operations using the ModelNet40 dataset, strictly adhering to the preprocessing protocol outlined in the paper (Section H), including zero-mean/unit-variance normalization and fixed set cardinality (n = 100 and n = 1000).
> >
> > The best performance we obtained by DeepSet (0.80 +/- 0.02 for n = 100 and 0.84 +/- 0.02  for n = 1000) closely matched the results originally reported by Zaheer et al. (0.82 +/- 0.02, 0.87 +/- 0.01 respectively).
> >
> > We then implemented an analogous model in which the equivariant DeepSets layers were replaced with QUANN-based equivariant layers as defined in Equation 14 of the manuscript. While the equivariant QUANN-1 achieved performance comparable to that of equivariant DeepSets, the equivariant QUANN-2 model significantly outperformed both. All the results can be found in Section D.1 of the latest revision.
> >
> > **2. The LAF aggregation experiments in Table 4 and 5 are not convincing. In principle, LAF can learn sum, mean, max, and other permutation invariant operations. For example in Table 4, it is surprising to me that DeepSets achieve an accuracy of 0.524 while LAF performs much worse at 0.156.**
> >
> > While LAF indeed performs worse than DeepSets in the Omniglot transfer tasks, the results highlighted by the reviewer correspond specifically to transfer learning, and therefore only indicate that the representations learned by LAF transfer less effectively to non-set tasks than those learned by DeepSets.
> >
> > Regarding the Omniglot set-classification task itself, LAF uses learnable power-transformation followed by sum as the core aggregation mechanism. We hypothesize that power-mean and power-sum aggregations, while capable of approximating various arithmetic compositions, may not provide an optimal aggregation geometry for this particular task. Potentially explaining the weaker performance LAD and HPDS (which uses learnable power-mean) relative to methods with more flexible or better-structured aggregation operators in this task.
> >
> > **3. (minor) The additional experiments on QM9 are a good addition but I wouldn't call it a real world dataset as its properties are obtained from simulations (see the Background and Summary sections of [1] or https://www.tensorflow.org/datasets/catalog/qm9), although they are close to experimental accuracy. Moreover, QM9 is a subset of GDB-11, which was artificially generated to enumerate all possible molecules containing up to 11 atoms of H, C, N, O, and F**
> >
> > Thank you for pointing this out.  We agree that this provides important context and we will add these details into the revised description of the experiment in Section F.1.

---

> > > ### Comment · Reviewer_3i4p · 2025-11-27
> > > **Response**
> > >
> > > Thanks for the extra efforts - I am raising my score to 6

---

### Official Review · Reviewer_ZtCS · 2025-10-22

**Soundness:** 4
**Presentation:** 3
**Contribution:** 3
**Rating:** 6
**Confidence:** 4

**Summary:**

This paper presents Universal Set Transformer (UST), which combines a set transformer with mini-batch consistency (MBC), resulting in a methodology that is both expressive and has the ability to handle large sets. Set and elements can be represented, performance is often better than baselines, and explainability is possible by means of element-wise attention scores.

**Strengths:**

The formulation of element-wise representation into the multiset attention is theoretically interesting. The integration of this multiset attention into an MBC processing, is practically relevant. The experimental results are convincing, the performance is good across tasks, the stability w.r.t mini-batch size is good.

**Weaknesses:**

The results and discussion read as if there are only advantages of the proposed method. What are limitations, where does UST not perform well? There is a discussion section, but it does not reflect on the possible disadvantages of UST.

Multiset attention is here posed as a completely new thing. In "Unlocking Slot Attention by Changing Optimal Transport Costs" [1] there is also attention across the elements in the multiset. The reference is missing in the related work.

**Questions:**

What is the difference between the multiset attention as presented in this paper, compared to ref [1] (see above)? If this distinction is significant, and can be motivated in technical/math terms, then I am willing to increase my score, because it is a good paper but this remaining point needs clarification.

---

> ### Comment · Reviewer_ZtCS · 2025-11-13
> **Review updated**
>
> This review was updated. I made a mistake and switched the review texts for two papers about sets that were in my batch. The other set paper is also corrected.

---

> ### Author Response · Authors · 2025-11-21
> **Response to Reviewer ZtCS**
>
> **1. The ablation is not clear, a discussion of various (left out / alternative) components and their impact on the performance is lacking.**
>
> QUANNs implement newly introduced neuralized Kolmogorov mean (NKM) as a form of learnable aggregation induced via an invertible neural function \psi.  In Section 5.3, we argue that NKM effectively acts as an inner regularization and prevents dimensional collapse, thus improving the learning process.  We assert benefits of NKM prepared three ablated models by: removing NKM completely (replacing \psi by identity function) (Ablation 1), removing NKM but keeping overall capacity of the model (Ablation 2), replacing normalization term in Eq. 4 (Ablation 4). These findings demonstrate that the inclusion of NKM and its invertible neural components is crucial for accurately modeling complex set functions.
>
> **2. In Table 3, it is not clear whether the numbers show that the method is capable of the task. The numbers vary greatly across tasks, which makes it hard to assess what the meaning of 'good' performance is here.**
>
> As these experiments independently approximate different aggregation functions, their numerical results naturally vary. Importantly, QUANN-1 outperforms all three ablated models in 9 out of 10 tasks—significantly so in 7 of them.
>
> **3. In related work, it is not clear to me how the methods from 3.2 are different from 3.1 - I know they are different, but a short explanation would make this section stronger. Regarding slot attention, a reference is missing, Unlocking Slot Attention by Changing Optimal Transport Costs [1]. Because they also address set prediction tasks, it's interesting to see how the proposed method compares to this reference.**
>
> We have renamed Section 3.2 to “Non-Janossy Methods” and added a preamble explaining the conceptual distinction between Janossy-based approaches and methods that achieve permutation invariance via non-Janossy strategies (lines 151-154). We also included the suggested reference, Unlocking Slot Attention by Changing Optimal Transport Costs.
>
> This method is primarily to decompose inputs (e.g., images) into latent slots for object discovery, rather than supervised permutation-invariant set function approximation. Adapting it for this task would require substantial architectural changes (e.g., removing spatial encoders and defining a slot-to-pooled readout), preventing fair comparison. Instead, we expanded our baselines by including LAF introduced in (Pellegrini et al., 2021).
>
> **4. In the experiments, it does not say whether there were tasks with sets of varying sizes within one task. Dealing with variable-sized sets would make the experiments more interesting.**
>
> In all experiments we used varying size sets. Details on this are provided in the Section F.1.
>
> **5. There is a discussion but it lacks a critical view of the learnable aggregator, when it may not be beneficial, etc.**
>
> We have extended the Discussion – Limitations section to provide a more critical perspective on learnable aggregators, including scenarios where they may offer limited benefit, potential overfitting risks, and situations where simpler fixed aggregators might be preferable (lines 463-471).
>
> **6. Can you discuss the various contributions of components that are reported in the ablation, why they matter, and their impact on the performance?**
>
> See our response to no. 1.
>
>
> **7. Can you explain the difference between the methods from 3.1 and 3.2?**
>
> Methods in Section 3.1 follow the factorization described in Equation 2, i.e., Janossy pooling. In contrast, methods in Section 3.2 do not admit a strict Janossy factorization; instead, they achieve permutation invariance through mechanisms such as slot attention or by learning an optimal ordering of the input set. To clarify this distinction, we renamed Section 3.2 to “Non-Janossy Methods” and added a brief preamble explaining the conceptual difference.
>
> **8. Do some of the experiments contain a task with variable-sized sets?**
>
> All our experiments involve tasks with variable-sized sets. For more details please see Supplementary Section F.1.
>
> **9. What do the numbers in Table 3 tell us, given that they are so broad in their range?**
>
> Please see our answer to point no. 2
>
> **10. Are there cases where it is not beneficial to learn the aggregator, but to fix it with a well-chosen function?**
>
> Yes, there are situations where a fixed aggregator may be preferable. For instance, if the target set function is known or expected to be additive or expansive, a simple sum naturally captures this structure. Similarly, when data are limited, a learnable aggregator may be unnecessarily flexible and prone to overfitting, whereas a fixed function provides useful inductive bias and computational simplicity. We have added this discussion to the Discussion – Limitations section (lines 463-471).

---

> > ### Comment · Reviewer_ZtCS · 2025-11-22
> > **Response to rebuttal**
> >
> > Dear authors, I am satisfied with the responses to my questions and concerns, the explanations and improvements to the paper are satisfactory, therefore I have increased my score to a 6.

---

### Official Review · Reviewer_71eq · 2025-11-01

**Soundness:** 3
**Presentation:** 3
**Contribution:** 3
**Rating:** 6
**Confidence:** 4

**Summary:**

The paper presents a novel approach for learning the pooling operator in set function approximation. By leveraging a neuralized version of the Kolmogorov mean, it manages to approximate various measures of central tendency.

**Strengths:**

The paper is well written and clear. The choice of Kolmogorov mean and its neutralized version is well-motivated. There is a sound discussion of the theoretical advantage of the proposed solution over alternatives, complemented with promising experimental results.

**Weaknesses:**

There were few works that explicitly addressed the problem of learning pooling operators in the past:

- Euan Ong, Petar Veličković, Learnable Commutative Monoids for Graph Neural Networks, LOG 2022.
- P. Zuidberg Dos Martires, Neural Semirings, NeSy 2021.
- G Pellegrini, A Tibo, P Frasconi, A Passerini, M Jaeger, Learning aggregation functions, IJCAI 2021.

While none of them directly suggests using the Kolmogorov mean, they all attempt to go beyond predefined aggregators, and at least one (neural semiring, arguably a not very popular paper) explicitly mentions the use of invertible neural networks. Positioning the contribution with respect to these works would better clarify its novelty.

Countability assumption. This is not really a weakness (most existing results, starting from deep sets, assume countable sets), but given its relevance (sets of real vectors are uncountable) it would be worth discussing the assumption in the limitations.

Minor:
- please check your citation reference fornatting and use citep when appropriate.
- Pk(X) -> Sk(X) [or viceversa] in the description of Eq. 2

**Questions:**

Can you clarify the main differences wrt alternative approaches for learning pooling operators?

Can you discuss more clearly the implications of the countability assumption?

---

> ### Author Response · Authors · 2025-11-21
> **Response to Reviewer 71eq**
>
> **1. There were few works that explicitly addressed the problem of learning pooling operators in the past:**
>
>   - Euan Ong, Petar Veličković, Learnable Commutative Monoids for Graph Neural Networks, LOG 2022.**
>
>   - P. Zuidberg Dos Martires, Neural Semirings, NeSy 2021.**
>
>   - G Pellegrini, A Tibo, P Frasconi, A Passerini, M Jaeger, Learning aggregation functions, IJCAI 2021.**
>
> **While none of them directly suggests using the Kolmogorov mean, they all attempt to go beyond predefined aggregators, and at least one (neural semiring, arguably a not very popular paper) explicitly mentions the use of invertible neural networks. Positioning the contribution with respect to these works would better clarify its novelty.**
>
> Thank you for bringing these works into our attention. We refer to these works in the revised manuscript across the Related work and Discussion sections.
>
> (a) Neural Semirings (Zuidberg Dos Martires, 2021).
>
> Neural semirings aim to “neuralize’’ algebraic structures by learning binary operations that satisfy (approximate) semiring axioms. Our work is fundamentally different because Kolmogorov means are not binary operations: they are set functions defined for unordered sets of arbitrary cardinality.
>
> While the special case of a 2-element set {x,y} can be written as: f^(-1)[f(x) + f(y)], which resembles the additive part of a semiring induced by f. This correspondence breaks down for larger sets, where Kolmogorov means act as nonlinear expectations rather than binary algebraic operators.
>
> Moreover, neural semirings are purely theoretical and were not demonstrated on set-learning tasks; they therefore cannot serve as a baseline. Our contribution is complementary: we learn monotone/invertible transforms defining Kolmogorov means and show they significantly improve performance on standard set-learning benchmarks.
>
> (b) Learnable Commutative Monoids (Ong & Veličković, 2022).
>
> LCMs learn a commutative binary operator​, used specifically for local message aggregation between pairs of adjacent nodes in a graph. Thus, their goal is to learn an algebraic operator for graph message passing between binary pairs of adjacent nodes.
>
> In contrast, we address global permutation-invariant aggregation over entire sets. Kolmogorov means to define a structured family of nonlinear expectations that operate on arbitrary-sized unordered sets, not binary pairs. Informally, LCMs aim to “learn the algebra’’ of local combination rules, while we aim to “learn the geometry underlying the algebra’. This difference in scope makes LCMs complementary but not comparable baselines.
>
> (c) Learning Aggregation Functions (Pellegrini et al., 2021).
>
> This work introduces a learnable aggregator (LAF) in which the pooling operator is constructed as a fixed-form combination of four independent power means—each of which is a special case of a Kolmogorov mean. While this provides some flexibility, the aggregation family is still structurally limited by the fixed power-mean architecture.  We have added LAF as an additional baseline in our experiments, and empirically it performs substantially worse than our method.
>
> **2. Countability assumption. This is not really a weakness (most existing results, starting from deep sets, assume countable sets), but given its relevance (sets of real vectors are uncountable) it would be worth discussing the assumption in the limitations.**
>
> We added this into the revised Discussion section - Limitations (lines 472-476).
>
> **3. Please check your citation reference formatting and use citep when appropriate.**
>
> All references were corrected using \citep and \citet when appropriate.
>
> **4. Pk(X) -> Sk(X) [or viceversa] in the description of Eq. 2**
>
> The typo in Equation (2) was corrected to be consistent with Equation 4.
>
>
> **5. Can you clarify the main differences wrt alternative approaches for learning pooling operators?**
>
> Besides the works discussed in Response 1, the only approaches we are aware of that explicitly learn a pooling operator are HPDS (Kimura et al., 2024) and Slot Attention (Locatello et al., 2020). Both differ from QUANNs in fundamental ways:
>
> (a) HPDS (Hölder Power Deep Sets)
>
> HPDS parameterizes the pooling operation using a generalized Hölder (power) mean, which is a special case of Kolmogorov mean. Although this yields a learnable pooling operator, its expressivity is inherently limited because the pooling is governed by just two parameters (the exponent and scaling).
>
> (b) Slot Attention
>
> The aggregation in Slot Attention is effectively a weighted arithmetic mean, where the weights are produced by the attention mechanism. While this produces a flexible soft clustering prior to pooling, the pooling operation itself is still structurally constrained to a weighted mean of the original values.
>
> **6. Can you discuss more clearly the implications of the countability assumption?**
>
> Please see our answer in no. 2.

---

### Official Review · Reviewer_oz2P · 2025-11-09

**Soundness:** 3
**Presentation:** 3
**Contribution:** 3
**Rating:** 6
**Confidence:** 3

**Summary:**

This paper introduces quasi-arithmetic neural networks (QUANNs) for learning set functions. By parameterizing the Kolmogorov mean function (also called the quasi-arithmetic mean, and hence the name of the proposed architecture), QUANNs can be seen as an extension of the Normalized DeepSets architecture, where the standard mean function in Normalized DeepSets is replaced with a learnable Kolmogorov mean. More generally, QUANNs further allow aggregation over subsets of set elements akin to the Janossy pooling. The authors provide explanation for the theoretical benefits QUANNs, as well as hypotheses for the practical advantage of QUANNs. The author verify their hypotheses by carrying out a set well designed experiments using both synthetic and real-world data. Empirically, QUANNs are shown to outperform state-of-the-art set neural network baselines.

**Strengths:**

The idea to parameterize the Kolmogorov mean function and replace the standard mean pooling in existing set neural network architectures is intuitive and interesting. Empirically, the resulting QUANN architecture outperforms all baselines. It is good to see that a simple and principled modification leads to effective performance gain.

The experiments are well designed to empirically verify the authors' hypotheses on the practical advantages of QUANNs.

**Weaknesses:**

Writing can be improved.
- Section 5.2 has a lot of references to the theorems in the appendix. It is difficult to follow the discussion in this section without having to read the theorems in the appendix. I think it would be helpful to at least state some informal and short versions of the theorems here.
- Please use citet{…} and citep{…} appropriately, e.g., use citep{…} for references that appear in lines 28-32. There are many misuses of citet{…} or cite{…} throughout the entire paper, please fix these. In Line 116, should define S_k(X) here, not P_k(X). Be consistent in the use of notations in Equation (2) and (4).


I am not sure how practical the new architecture is. The authors should discuss the practical relevancy of set representation learning in the current state of machine learning. Can't LLMs/vLLMs solve the tasks considered in the experiments?

**Questions:**

For the synthetic data experiment, Ablation 2 should in theory be strictly better than Ablation 1. Why do the results in Table 3 show the opposite? Was it because of bad training, bad tuning/regulations, or else?

Also for the synthetic data experiment, many of the functions involve computing some average in one way or another. This aligns with the normalization used in QUANN. What happens if you want to learn functions that are not averages, but sums (e.g. vector norms)?

---

> ### Author Response · Authors · 2025-11-21
> **Response to Reviewer oz2P**
>
> **1. Section 5.2 has a lot of references to the theorems in the appendix. It is difficult to follow the discussion in this section without having to read the theorems in the appendix. I think it would be helpful to at least state some informal and short versions of the theorems here**
>
> We incorporated the Universality Approximation Theorem for QUANNs (Theorem 5.1) into the main text of the revised manuscript (lines 223-227). However, due to space limitations, the remaining propositions are retained in the Appendix. To improve clarity and readability, we also added informal proposition names alongside their reference (Section 5.2).
>
> **2. Please use citet{…} and citep{…} appropriately, e.g., use citep{…} for references that appear in lines 28-32. There are many misuses of citet{…} or cite{…} throughout the entire paper, please fix these. In Line 116, should define S_k(X) here, not P_k(X). Be consistent in the use of notations in Equation (2) and (4).**
>
> All references were corrected using \citep and \citet when appropriate and the typo in Equation (2) was corrected to be consistent with Equation 4.
>
> **3. I am not sure how practical the new architecture is. The authors should discuss the practical relevancy of set representation learning in the current state of machine learning. Can't LLMs/vLLMs solve the tasks considered in the experiments?**
>
> We have added a paragraph in the revised Discussion highlighting some of the practical applications of QUANNs (lines 478-485). Regarding the reviewer’s question, we note that LLMs/vLLMs are not well-suited for the set function approximation tasks considered in our experiments, because they operate on ordered token sequences and are fundamentally not permutation invariant.
>
> **4. For the synthetic data experiment, Ablation 2 should in theory be strictly better than Ablation 1. Why do the results in Table 3 show the opposite? Was it because of bad training, bad tuning/regulations, or else?**
>
> In Section 5.3, we argue that allocating part of the model’s capacity to the invertible function \psi effectively acts as an inner regularization, improving the learning process. Without this invertible component, Ablation 2 becomes over-capacitated and prone to overfitting, leading to worse generalization compared to the smaller Ablation 1.
>
> **5. Also for the synthetic data experiment, many of the functions involve computing some average in one way or another. This aligns with the normalization used in QUANN. What happens if you want to learn functions that are not averages, but sums (e.g. vector norms)?**
>
> Of the ten aggregating functions used in the synthetic data experiment, only four can be characterized as central tendencies (or informally as types of average), while four others are measure of extremes (including maximum of the vector norm), and additional two are higher moments (cf Table 7 of the manuscript).  QUANN-1 outperformed ablations in all but one of these – skewness. Nonetheless,  aggregates of expansive nature, such as SUM, are expected to be more challenging for QUANN - which we explain in the Section 5.2 - Sum decomposition. We also discuss this as a key limitation of our method in the revised Discussion - Limitations section (lines 463-471).

---

### Author Response · Authors · 2025-11-21
**General comment.**

We would like to thank all reviewers for their thorough evaluation of our work and for the insightful comments and suggestions. In response to the feedback, we have revised the manuscript and extended our experimental evaluation by adding an additional baseline, LAF (Pellegrini et al. 2021), and by incorporating experiments on the real-world dataset - QM9 (Ramakrishnan et al. 2014}.

Below, we address each reviewer’s comments and questions in detail.

---

### Meta-Review · Area_Chair_guYY · 2025-12-15

**Summary:**

The paper introduces Quasi-Arithmetic Neural Networks (QUANNs), which utilize a novel trainable pooling mechanism called the Neuralized Kolmogorov Mean (NKM). This approach addresses the limitations of fixed pooling operations (e.g., sum, max) inherent in standard set-learning architectures like DeepSets and PointNet. The authors provide a theoretical analysis that proves QUANNs are universal approximators for a broad class of set functions and ensure permutation invariance. Empirically, the method is shown to outperform state-of-the-art baselines on various benchmarks.

**Reviewer Concerns:**

- Multiple reviewers initially raised concerns that the paper lacked comparisons to other learnable aggregation functions (e.g., LAF, PNA) and questioned the novelty relative to these works

- One reviewer noted a significant discrepancy between the reported DeepSets accuracy on ModelNet40 and the original paper's results

- Another reviewer questioned the practical relevance of set representation in the age of LLMs and found the ablation study results (where a larger model performed worse) counterintuitive

**Reviewer Scores:**

The authors provided a comprehensive rebuttal that effectively addressed the major concerns, resulting in a positive trajectory of scores during the discussion period.

Importantly, they added the LAF baseline (showing QUANN superiority) and conducted new experiments replicating the exact preprocessing of the original DeepSets paper to resolve the accuracy discrepancy. The authors also clarified the positioning against related work (Neural Semirings, etc.), as well as variable set sizes and the role of the learnable aggregator.

Overall, AC believes that if a full discussion period continued, the scores would likely remain stable at this consensus of 6 as the critical flaws identified (missing baselines/discrepancies) were empirically resolved.

---

### Decision · Program_Chairs · 2026-01-26

Accept (Poster)